# Assessing the effect of lithological setting, block characteristic and slope topography on the runout length of rockfalls in the Alps and on the La Réunion island

Kerstin Wegner[1], Florian Haas[1], Tobias Heckmann[1], Anne Mangeney[2], Virginie Durand[2,a], Nicolas Villeneuve[2,3], Philippe Kowalski[2,4], Aline Peltier[2,4], Michael Becht[1]

[1]Chair of Physical Geography, Catholic University of Eichstaett-Ingolstadt, Eichstaett, 85072, Germany
[2]Université de Paris, Institut de Physique du Globe de Paris, CNRS, F-75005 Paris, France
[3]Université de La Réunion, Laboratoire Géosciences Réunion, F-97744 Saint Denis, France
[4]Observatoire Volcanologique du Piton de la Fournaise, Institut de Physique du Globe de Paris, F-97418 La Plaine des Cafres, France
[a]now at: Helmholtz Centre Potsdam German Research Centre for Geosciences, GFZ, 14473 Potsdam, Germany

*Correspondence to*: Kerstin Wegner (KWegner@ku.de)

**Abstract.**

In four study areas within different lithological settings and rockfall activity, LiDAR data was applied for a morphometric analysis of block sizes, block shapes and talus cone characteristics. This information was used to investigate the dependencies between block size and block shape and lithology on the one hand and runout distances on the other hand. In our study, we were able to show that lithology seems to have an influence on block size and shape and that gravitational sorting did not occur on all of the studied debris cones, but that other parameters apparently control the runout length of boulders. Such parameter seems to be the block shape, as it plays the role of a moderating parameter in two of the four study sites, while we could not confirm this for our other study sites. We also investigated the influence of terrain parameters such as slope inclination, profile curvature and roughness. The derived roughness values show a clear difference between the four study sites and seem to be a good proxy for block size distribution on the talus cones and thus could be used in further studies to analyze a larger sample of block size distribution on talus cones in different lithology.

## 1. Introduction

Rockfall is an important geomorphic process on steep rock slopes and thus plays a significant role for the geomorphic dynamics especially in high mountainous regions (e.g. Hungr and Evans, 1988, Krautblatter and Dikau, 2007, Bennett, et al., 2012, Frattini et al., 2012). Rock fragments are detached from cliff faces (primary rockfall) or remobilized from sediment stores (secondary rockfall) downslope (e.g. Rapp 1960, Krautblatter and Dikau, 2007), move in a combination of falling, bouncing, rolling or sliding (e.g. Luckman, 2013a, Crosta et al., 2015) and are subsequently deposited on storage landforms such as talus cones. Even though rockfalls often take place in remote areas, they can still pose a potential natural hazard and can cause

damage to human lives and local infrastructure facilities (e.g. Pfeiffer and Bowen, 1989, Dorren, 2003, Ravanel et al., 2010, Volkwein et al., 2011, Frattini et al., 2012, Heiser et al., 2017).

The occurrence and magnitude of rockfall depend on the preconditioning and the preparatory factors (e.g. Meißl, 1998, Dorren, 2003, Dietze et al., 2017a) as well as on triggering events. The preconditioning and preparatory factors are mainly: lithology, topography of the slope (aspect, steepness, altitude), vegetation (e.g. Meißl, 1998, Jaboyedoff and Derron, 2005), rainfall and weathering, frequency of freeze/thaw cycles, sun exposure and root growth (e.g. Meißl, 1998, Jaboyedoff and Derron, 2005, Krautblatter and Dikau, 2007, Frattini et al., 2012, Crosta et al., 2015). Triggering events include volcanic or seismic forcing (Hibert et al., 2017, Durand et al., 2018).

Due to the change in the settlement structure in mountain regions, but also in the context of global climate change, many studies exist about rockfall processes, focusing on the modelling of runout trajectories and the prediction of rockfall events (e.g. Kirkby and Statham, 1975, Meißl, 1998, Agliardi and Crosta, 2003, Dorren 2003, Copons et al., 2009, Jaboyedoff and Labiouse, 2011, Frattini et al., 2012, Nappi et al., 2013, Wichmann, 2017, Volkwein et al., 2018, Caviezel et al., 2019), as well as on the measurement of rockfall activity by seismic monitoring (e.g. Vilajosana et al., 2008, Hibert et al., 2011, Farin et al., 2015, Dietze et al., 2017a, Dietze et al., 2017b, Durand et al., 2018, Feng et al., 2019). Most of these studies use LiDAR (light detection and ranging) techniques such as airborne laser scanning (ALS) as well as terrestrial laser scanning (TLS) to improve the understanding of this geomorphic process (e.g. Jaboyedoff et al., 2007, Abellán et al., 2011, Haas et al., 2012, Heckmann et al., 2012, Royán et al., 2014, Strunden et al., 2015, Sala et al., 2019). In recent times Structure-from-Motion (SfM) is also increasingly used for such kind of studies (e.g. Kromer et al. 2019, Vanneschi et al., 2019, Guerin et al., 2020).

In the context of hazard assessment, but also for geomorphological models, not only the transported volumes, but also the analysis of the maximum runout distance of blocks plays an important role especially in populated mountain regions (e.g. Jaboyedoff and Labiouse, 2011, Volkwein et al., 2011, Lambert et al., 2013, Caviezel et al., 2019). Factors influencing the runout distance and the trajectory of blocks include topographic conditions, like the slope inclination, height of the fall and curvature conditions (e.g. Meißl, 1998, Frattini et al., 2012, Leine et al., 2014). Other parameters are the surface properties including protection measures and the roughness of the slope (e.g. Meißl, 1998, Leine et al., 2014, Gratchev and Saeidi, 2019). Some studies show the influence of the properties of the rockfall itself, like the volume, geology, and size and shape of the blocks on the runout distances (e.g. Kirkby and Statham, 1975, Meißl, 1998, Leine et al., 2014). In addition, the kinematical properties of the blocks and the restitution parameter also influence the depositioning (e.g. Azzoni and de Freitas, 1995, Jaboyedoff and Labiouse, 2011, Ji et al., 2019, Sandeep et al., 2020). The size and shape of blocks and their interaction with slope properties are considered to be an important factor for the travel distance (e.g. Pfeiffer and Bowen, 1989, Meißl, 1998, Frattini et al., 2012, Leine et al., 2014).

The influence of block shape, block size and slope topography on the runout distance was investigated only by a rare number of studies (Azzoni and de Freitas, 1995, Haas et al., 2012, Fityus et al., 2013) and mainly in case studies or tests under laboratory conditions (Okura et al., 2000, Glover et al., 2015a, Cui et al., 2017, Wang et al., 2018, Gratchev and Saeidi, 2019). Haas et al. (2012) already stated in their case study that the influence of the lithology on block shape and block size must be

investigated with a broader view. This could be done e.g. in areas with different lithological settings, as the size and shape of rock fragments are determined by their lithological properties, among other influencing factors (Haas et al., 2012, Fityus et al., 2013). Some studies described the deposition of boulders on the talus cone as gravitational sorting, where larger blocks are deposited in the distal part and smaller blocks at the upper part of the talus slope (e.g. Statham, 1973, White, 1981, Whitehouse and McSaveney, 1983, Kotarba and Strömquist, 1984, Jomelli and Francou, 2000, Sanders et al., 2009, Messenzehl and Dikau,

2017, Popescu et al., 2017, Kenner, 2019). The gravitational sorting of the blocks on the slope is a key factor for the roughness component of talus slopes and thus on the runout length of following rockfall events (Hungr and Evans, 1988), indicating a potential feedback loop in the formation of talus landforms.

The aim of this study is to carry out a comparative investigation of the morphometric properties and runout distances of rockfall fragments in mountain regions within different lithological settings. For our analysis we selected four sites with different

lithological conditions, different rockfall activity and existing data with a very high quality. Due to lithological differences of the cliffs, we expected different statistical distributions of block sizes and block shapes on the talus slopes. The study is conducted using high-resolution digital terrain models (DTMs) and point clouds created from TLS surveys. Based on the research of Haas et al. (2012), we determined different block properties, size and shape, and analyzed them in the context of runout distances and talus morphology.

**2. Study Sites**

Four areas in high mountainous regions were selected for this investigation. Three of these areas are situated in the Alps, one area is located on the island of La Réunion (Fig. 1). The areas differ mainly with regard to the lithological conditions.

All areas are characterized by a recent rockfall activity and a clearly distinguishable rock face with an associated scree slope. The selected study sites represent different processes such as rock topples and block fall. For Gampenalm (GA) and Dreitorspitze (DTS), the blocks are assigned to one rockfall event. For Piton de la Fournaise (PF) and Zwieselbach valley

(ZBT), the slope is characterized by deposited material from continuous rockfall processes, but also major events cannot be excluded. A further criterion for the selection of the area was that both the rock faces and the talus cones were clearly and completely visible to ensure a complete and dense LiDAR acquisition.

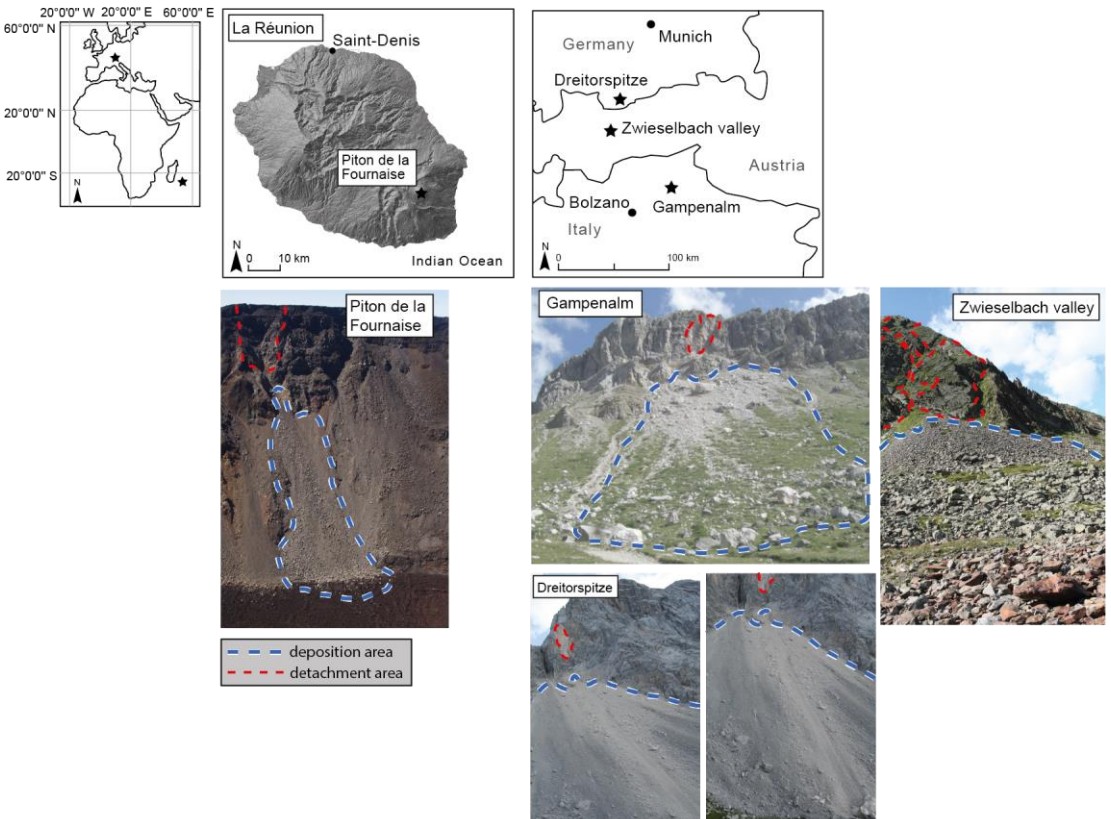


**Figure 1. Geographical location of the study sites located on La Réunion and in the European Alps. The stars locate the location of the study sites. The red dotted lines limit the detachment areas of the rockfall events. The blue and white dotted lines represent the deposition area. (Source of the overview base map: ESRI, HERE, Garmin, OpenStreetMap, contributors and the GIS user community).**


The areas GA and DTS are located in the southern and the northern Alps and consist of sedimentary rocks specifically of different limestone formations (Table 1). The GA area is located in the Dolomites and is dominated by the thick banked Rosengarten Dolomite (c.f. Haas et al., 2012). DTS is dominated by the thick banked Wetterstein limestone. In both areas, major rockfall events occurred in recent years.

The ZBT in the Stubaier Alps is located in the area of the crystalline Central Alps and is characterized by slated gneiss and metamorphic granites. Major rockfall events during the last years are not known, but the deposits on the talus cone indicate recent rockfall activity including also bigger blocks. For GA, DTS and ZBT the precondition and preparatory factors cannot be determined exactly, but thawing of mountain permafrost can be excluded due to the altitude and exposure. As we do not have temperature and precipitation data, we cannot give any information about this.


**Table 1. Morphological and lithological characteristics of all investigated talus slopes.**

| Study Area | PF | ZBT | GA | DTS |
|---|---|---|---|---|
| Altitude [m a.s.l.] | 2632 | 2278 | 2450 | 2682 |
| Lithology | Basalt | Gneisses, glimmers, metamorphic granits | Triassic dolomites, limestone, psephite | Triassic limestone, dolomite |
| **Mean slope inclination [°]** | | | | |
| talus cone | 36 | 29 | 32 | 32 |
| cliff | 55 | 50 | 59 | 66 |
| **Length of [m]** | | | | |
| talus cone | 250 | 55 | 250 | 300 |
| cliff | 150 | 60 | 150 | 100 |
| Mean annual precipitation [mm year$^{-1}$] | 3000–4250 | 1000 | 862.2 | 1500 |
| Mean annual temperature [°C] | 13.8 | 0.9 | 7.8 | 6.7 |
| Density [g/cm³] | 2.99 | gneiss: 2.80 granite: 2.64 | limestone: 2.55 dolomite: 2.70 | limestone: 2.55 dolomite: 2.70 |
| Friction angle φ [°] | 35–38 | gneiss: 26–29 | limestone: 31–37 | limestone: 31–37 |
| Tensile strength [MPa] | 10–30 | gneiss: 5–20 granite: 7–25 | limestone: 5–25 dolomite: 14 | limestone: 5–25 dolomite: 14 |

The test site PF (Dolomieu crater) on La Réunion is the only area outside the Alps and is located in the Indian Ocean, east of Africa, but politically it belongs to France as an overseas department. PF is one of the most active volcanoes in the world (e.g. Peltier et al., 2009a). For the period from 1950 to 2013, 93 eruptions have been documented (Staudacher et al., 2016). Due to a summit collapse during an eruption in 2007 (e.g. Peltier et al., 2009b) a 340 m deep caldera was formed (e.g. Staudacher et al., 2016) on an area of 1100 x 800 m (e.g. Urai et al., 2007). On this volcano, the composition of the lava is mainly bimodal with a combination of aphyric basalts and olivine rich basalts (e.g. Peltier et al., 2009a, Lénat et al., 2012). Due to the high tectonic stress (e.g. Merle et al., 2010, Staudacher et al., 2016), the high volcanic activity that generate deformation and seismic


activity (e.g. Sens-Schönfelder et al., 2014, Peltier et al., 2018) and the layering of different lava flows, the rim is very unstable and thus prone to high rockfall activity (Hibert et al., 2017, Durand et al., 2018). This area certainly differs most clearly from all other studied areas. Besides the volcanic rocks, both the cliff and the talus cones are very young landforms, as the geomorphic forming started right after the emergence of the caldera in 2007. Further differences are the high deformation and seismic activity and the extremely high precipitation. Seismicity and rainfall have been show to play a role in rockfall triggering (Hibert et al., 2017, Durand et al., 2018), but should not have any influence on the runout lengths of single blocks, so that for the present investigations primarily the differences in lithology and in the case of PF the age have to be considered.

## 3. Materials and methods

### 3.1 Data acquisition and processing (TLS)

The data of all study sites have been acquired with a terrestrial 3D long range laser scanner. Two systems were used: the Riegl LMS-Z420i and the Riegl VZ-4000. Each scanner works on the same principle of time of flight, but due to laser configurations, the scanning distance of the VZ-4000 is four times longer with 4000 m compared to 1000 m. Both systems provide colour information due to integrated camera systems in order to colorize the point clouds. The RGB values of the pictures allow the filtering of vegetation in the pre-processing of the data and simplify the visualisation of e.g. individual blocks. All important technical information of both devices is listed in Table 2.

Due to the long distance between scanner and target and poor reflectance of the volcanic material in the Dolomieu crater, we used the VZ-4000 for this study site. All other test sites were surveyed using the LMS-Z420i. To minimize shadowing effects, several scan positions were necessary at each site, which had to be referenced using manual adjustment and ICP algorithms (Table 3). Both tools are implemented in the software RiScan Pro (v2.2.1 www.riegl.com).

**Table 2. Technical data of the two terrestrial laser scanning systems Riegl LMS-Z420i and VZ-4000 (RIEGL Laser Measurement Systems GmbH, 2010, RIEGL Laser Measurement Systems GmbH, 2020). The values of the VZ-4000 refer to measurements at a rate of 30 kHz.**

|  | LMS-Z420i | VZ-4000 |
|---|---|---|
| Max. measurement range | 1000 m | 4000 m |
| Measurement rate | 8000 pts./sec. | 23000 pts./sec. |
| Accuracy | 10 mm | 15 mm |
| Precision | 4–8 mm | 10 mm |
| Laser wavelength | Near infrared | Near infrared |
| Laser beam divergence | 0.25 mrad | 0.15 mrad |

After the referencing, the data were exported as ASCII files containing *x*, *y*, and *z* coordinates as well as RGB values for further analysis in SAGA GIS/LIS (Conrad et al., 2015; Laser Information System LIS: www.laserdata.at). We worked on the vegetation filtered and homogeneously thinned point clouds and created digital terrain models (DTMs) for all test sites with a raster resolution of 0.75 m using the lowest *z* value. Table 3 provides information about each study site including the vertical and horizontal scan resolution, the referencing precision, number of points in the raw data set and the point density (pts /m²).

**Table 3. Information about the TLS surveys for each study area.**

|  | TLS system | Scan resolution | Referencing precision | Number of points raw data set | Point density (pts/m²) |
|---|---|---|---|---|---|
| **PF** | VZ-4000 | 0.02° | 0.007 – 0.013 m | >60e+6 points | 55 pts/m² |
| **ZBT** | LMS-Z420i | 0.05° – 0.1° | 0.018 – 0.060 m | 1.5e+6 points | 50 pts/m² |
| **GA** | LMS-Z420i | 0.05° – 0.1° | 0.012 – 0.058 m | 8e+6 points | 54 pts/m² |
| **DTS** | LMS-Z420i | 0.05° – 0.2° | 0.015 – 0.065 m | 3e+6 points | 54 pts/m² |

## 3.2 Determination of block size, block shape and runout length

As the block size and shape have a great influence on the runout distance of the falling rock material, we measured according to the workflow of Haas et al. (2012) the dimensions of the three axes (*a*, *b* and *c*) of single boulders for every study area (PF: n=255, GA: n=618, ZBT: n=65, DTS: n=182). We selected blocks that were deposited on the talus surface beginning from the upper to the lower parts of the talus cones. Based on the coloured LiDAR point clouds every block larger than approximately ~0.5 m was manually measured in the software RiScan Pro. Figure 2 shows an idealized sketch of a boulder with the three measured axes.

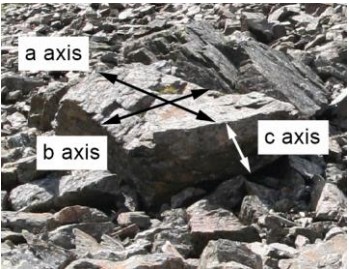

**Figure 2. Deposited block on the talus cone ZBT with the three measured dimensions of the axes *a*, *b*, and *c*.**

We followed the work of Haas et al. (2012), which used the formula of Valeton (1955) as an indicator of the block shape. By using the measured three axes, the volume (Eq. 1) and the axial ratio (Eq. 2) was approximated. The derived axial ratio of the

measured blocks describes their block shape. For the calculation of the axial ratio, the parameter of the axis *b* must be set to 1
(cf. Valeton, 1955). With this method, blocks with an axial ratio of 1 can be described as a cuboid or as equant, but the method
does not reflect the roundness of blocks. In the following we will also use the term of a small or low axial ratio. The larger the
value of the axial ratio, the more elongated or irregular shaped the blocks are. Here we will refer to a high axial ratio. The
different lithology of the study areas determines the shape of the deposited blocks to a certain degree as predisposition (Glover
et al., 2015b).


$$Block\ volume = a * b * c \tag{1}$$
$$Axial\ ratio = a / b / c \tag{2}$$

Since a 3D approach to measure the blocks would be too time-consuming for this number of blocks, we decided to use the
lengths of the axes of the blocks for all study sites. In order to be able to make an assessment about the overestimation of the
calculated block volume, we selected a sample of ten blocks with different sizes for each study area (Fig. 3). Using the TLS
point clouds, the block volume was derived in RiScan Pro. The comparison between volume estimation via the three measured
axes and the 3D volume calculation in Riscan Pro in all four areas shows that the volumes of the blocks are constantly
overestimated. It is also visible that this overestimation appears relatively small in the area ZBT, which can be explained by
the rather angular structure of the blocks (cf. Fig. 2). This provides an indication that the overestimation can be explained
primarily by the fact that the axis calculation assumes rectangular edges for the volume calculation, which is certainly not true
for all blocks of the other areas. However, in view of the uniform use of the axis method and the systematic overestimation in
all areas, it seems permissible to us to calculate the volume via this simple method and to use it for the further analyses.

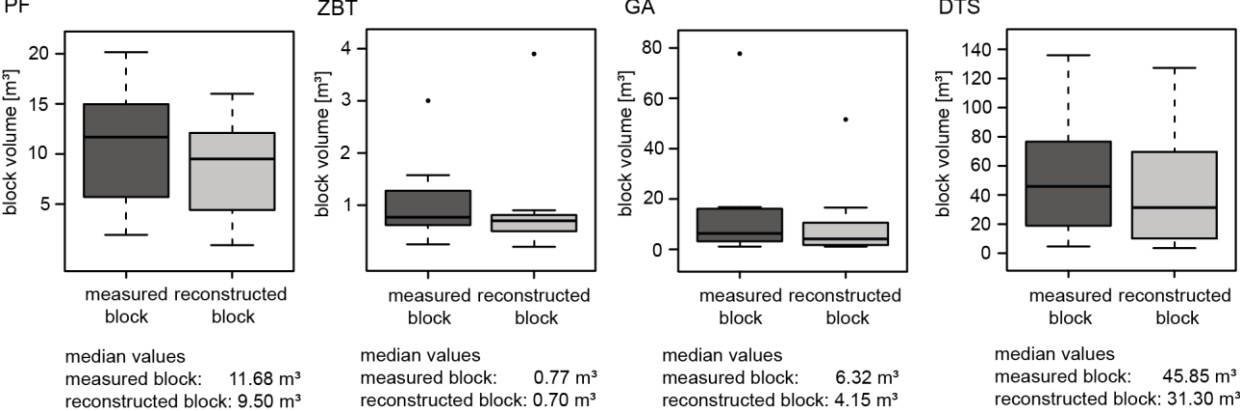


**Figure 3. Boxplots of measured and reconstructed block volume. The thick black lines within the boxes show the median bounded by the 1st quartile (25th percentile) and the 3rd quartile (75th percentile). The boxes show the interquartile range (IQR). The whiskers of the boxplots mark the minima and maxima of the data. The circles represent data which exceed the 1.5fold IQR.**

For the study areas of DTS and GA we determined the Euclidean distance from their detachment zone to each measured boulder to obtain the runout length. Since the exact detachment zone could not be determined at PF and ZBT, we measured the Euclidean distance from the beginning of the transition between cliff and talus cone to each boulder instead. In order to be able to compare the runout distances between the slopes of the study areas and as they differ greatly in length, we normalized the runout lengths for each talus cone to the interval [0,1]. We have assumed that a value of 0 stands for the detachment area or the area between the rock face and the slope. The value of 1 stands for the maximum runout length of a block.

### 3.3 Morphometric slope properties

### 3.3.1 Slope inclination and curvature

Based on the high-resolution DTMs of the study sites we performed a spatial analysis of the talus cones including morphometric properties with a presumed influence on the deposition (e.g. Wang and Lee, 2010, Frattini et al., 2012, Crosta et al., 2015) and runout distances of rockfall boulders (e.g. Glover et al., 2015a): slope inclination and profile curvature. The slope inclination was derived based on the DTMs according to Zevenbergen and Thorne (1987).

Additionally, we created three longitudinal profiles with a swath width of 10 m from the highest part of the cone to the distal boundary in order to characterize slope morphology of all four test sites to show changes in slope inclination and profile concavity (cf. Hergarten et al., 2014). We applied a kernel density estimation (cf. Cox, 2007) to compare the distributions of slope inclination between rock faces and talus cones, and between the study sites.

### 3.3.2 Surface roughness

The roughness of the slope can serve as a proxy for the particle size distribution on the talus cone. This parameter could give more information regarding gravitational sorting pattern where it indicates a coarsening towards the footslope and/or a concentration of coarse material at the foot of the rock wall. This is to determine and analyse the sorting of particle sizes beyond the measured block sizes of the single debris cones. For this purpose, we calculated the roughness based on the high-resolution TLS point clouds. The used algorithm fits a plane for each point of the point cloud in its local neighbourhood with a search radius of 3 m (SAGA Module Surface Roughness (PC), Laser Information System LIS, 2010, www.laserdata.at). Using the plane, the standard deviation of the distance to the fitted plane for each point was derived. To obtain the spatial distribution and the change of roughness along the slope for each study site, we classified the elevation values of the talus cones ($z$ value) into 10 m classes. In order to be able to establish comparability between the study areas, we normalized the height values of the point clouds. Furthermore, we classified the block volumes to check whether they correspond to the spatial distributed roughness of the deposited blocks on the talus cone.

## 4. Results and Discussion

### 4.1 Morphometric analyses

**4.1.1 Block size and block shapes**

In Table 4 we have listed mean and median values for block size and shape for the four study sites. To show the distribution of the block properties, size and shape, we have visualized the measured blocks with the longest axis greater than ~0.5 m in Figure 4 using boxplots. With regard to the block volumes, the four investigated sites differ, in some cases very significantly. The smallest block sizes (Table 4) can be found in the area ZBT with a median value of 0.08 m³ and the largest block sizes in

the area DTS with a median value of 13.16 m³. The median is below 4 m$^3$ in the three areas PF (3.38 m³), ZBT (0.08 m³) and GA (1.63 m³). The data shows a very homogeneous distribution and only small maximum values in PF and ZBT. Whereas in GA and DTS a very large dispersion and high maximum values (Table 4) are obvious. This indicates that a correlation between the lithological conditions and the block sizes involved in rockfall processes is very likely, as the largest blocks are mainly found in areas with banked limestones. This is clearly shown in the Wetterstein limestone and to a lesser degree in the Dolomite

area. The extremely small block sizes are particularly striking in the area ZBT, which consists of metamorphic rocks. This can be explained by the rather slated and thus platy rock structure of the gneisses and by the long tectonic history of these rocks (c.f. Fig. 2).

In contrast to the block sizes, the block shapes (Fig. 4) in all four areas show a high dispersion between equant and rather elongated or irregular shaped blocks. The most elongated blocks can be found in the ZBT area with a median value of 2.63

(Table 4), which again can be explained by the slated structure of the gneisses. The limestones of the areas GA and DTS are very similar in the median, mean block shape and the dispersion of the data (Table 4, Fig. 4). The lowest values and therefore the most cuboid shaped blocks were found in the area PF with a median value of 1.73 consisting of its volcanic rocks. Whether this is due to lithology or to tectonic stresses caused by the high seismic activity and the resulting fissures and cracks cannot be finally clarified with the data of this study.

With regard to the block shapes and block size distributions in the areas, however, it can be concluded that the investigated areas show sufficient differences in both block sizes and shapes, and that this is very likely a consequence of the lithological setting. This allows for a more detailed investigation of the relationships between block shape, block size and runout distance. We will also analyze how the different process activity affects the deposition and runout distance.

**Table 4. Mean and median values for the parameters block size and block shape for each study site. The values for block size and shape (axial ratio) were calculated using Eq. (1) and Eq. (2) with b = 1.**

|  | Block size | | Block shape | |
|---|---|---|---|---|
|  | Mean [m³] | Median [m³] | Mean | Median |
| **PF** | 5.07 | 3.38 | 2.00 | 1.73 |

| | | | | |
|---|---|---|---|---|
| **ZBT** | 0.29 | 0.08 | 2.87 | 2.63 |
| **GA** | 11.86 | 1.63 | 2.58 | 2.33 |
| **DTS** | 70.19 | 13.16 | 2.78 | 2.40 |

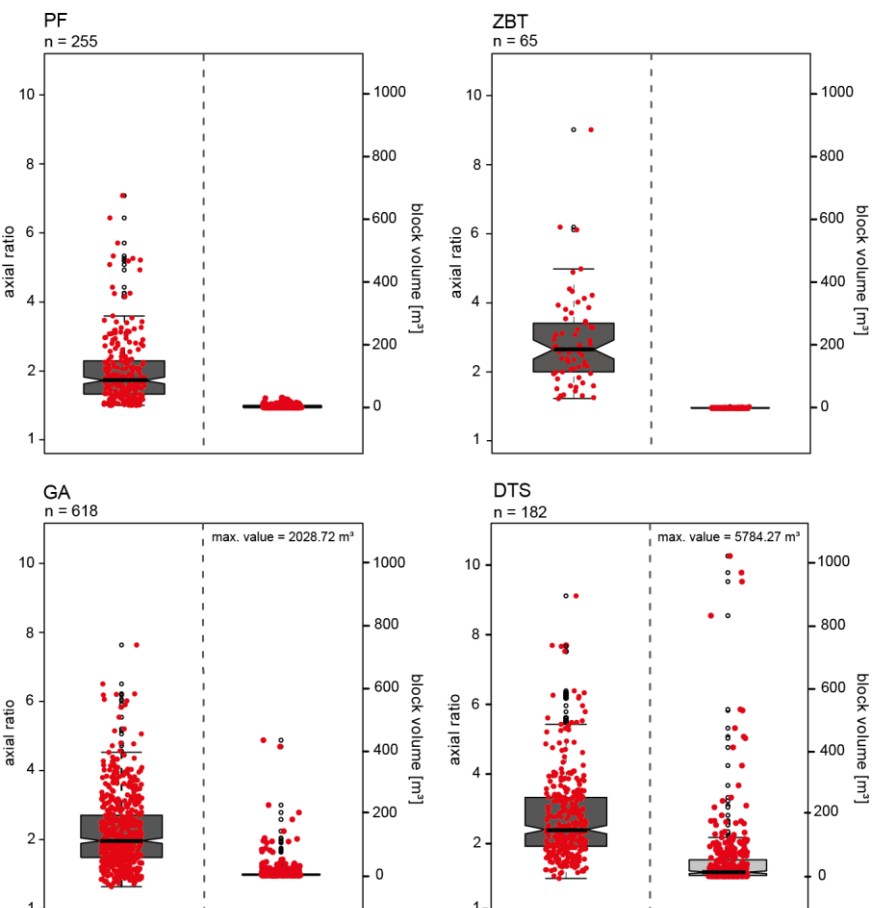

**Figure 4. Boxplots of volume and shapes for the measured blocks. The red coloured points correspond to each measured blocks. The thick black lines within the boxes show the median bounded by the 1st quartile (25th percentile) and the 3rd quartile (75th percentile). The boxes show the interquartile range (IQR). The whiskers of the boxplots mark the minima and maxima of the data. The circles represent data which exceed the 1.5fold IQR.**

### 4.1.2 Slopes of the talus cones

In addition to the block shape and block size, the topography of the talus cones plays also an important role and has to be considered for a runout analysis. Thus, we analyzed the following form parameters: slope inclination and curvature/slope profiles (Fig. 5, 6).

The average slope inclinations of the talus cones lie between 28° and 36° and thus within the range for such landforms (e.g. Pérez, 1989, Pérez, 1998, Francou and Manté, 1990, Jomelli and Francou, 2000, Sanders et al., 2009, Luckman, 2013b,

Popescu et al., 2017, Volkwein et al., 2018). The mean slope inclination values for the talus cones of our study sites are for PF 36°, ZBT 29°, GA 32° and DTS 32°. This can provide an indication of the dependence between rock material and slope inclination of gravel or blocky material and thus supports the findings of other studies.

**ZBT**

Kenner et al., (2019) reported in their study on amphibolite sediments slope angles of 35° and Messenzehl and Dikau (2017) described in their survey on metamorphic rocks slope angles of 34–36° and 33°–34°. Additionally, Gerber (1974) showed in his study slope inclination of 33° for gneisses and metamorphic granites. For the ZBT we found only 28.9° for the metamorphic rocks, which is significantly lower than in the cited studies. It must be noted that especially in ZBT the talus cone can be divided into two sectors, which differ very clearly in terms of slope (Fig. 5D). Thus, in the left area we find significantly higher slope inclinations with 43°, which are higher than the values from the literature, whereas in the right area rather significantly lower slope inclinations are found.

**PF**

The highest slope inclinations can be found on PF with 36.0° and its basaltic lava material. Yamamoto et al. (2005) conclude that basaltic material cannot be deposited on slopes of more than 33°. However, the higher average slope inclination at the PF compared to the findings of Yamamoto et al. (2005) can be explained by the rock surface conditions, which also play an important role, since fresh volcanic rocks in particular are characterized by a high micro-roughness, especially, where the lava could cool very quickly. Thus the slope seems to be controlled by the friction coefficient of the deposited material. In particular, for PF, analysis of seismic data for rockfalls suggest a friction coefficient of 35° for the talus slope (Hibert et al., 2011). This is supported by the distributions of the slope inclinations (c.f. Fig. 5C, 6C), where the data scatter is lowest in the PF area. Here the distribution is very peaked and differs clearly from the distributions of the other three areas. The shape of the slope profiles (Fig. 5A, 6A) also points in a similar direction. The talus cone of PF follows a straight line over the entire length of the slope, whereas the other cones show a slight convexity in the upper and middle slope and a basal concavity at the end of the slope, which is in good agreement with the works of Kotarba and Strömquist (1984), Luckman (2013b), and Popescu et al. (2017). Nevertheless, it is very likely that the talus cone of the PF represents a pure talus cone and is reshaped since then by persistent rockfalls, whereas the talus cones of the other areas represent much older forms where different types of geomorphic processes occur (e.g. debris flows, avalanche deposits).

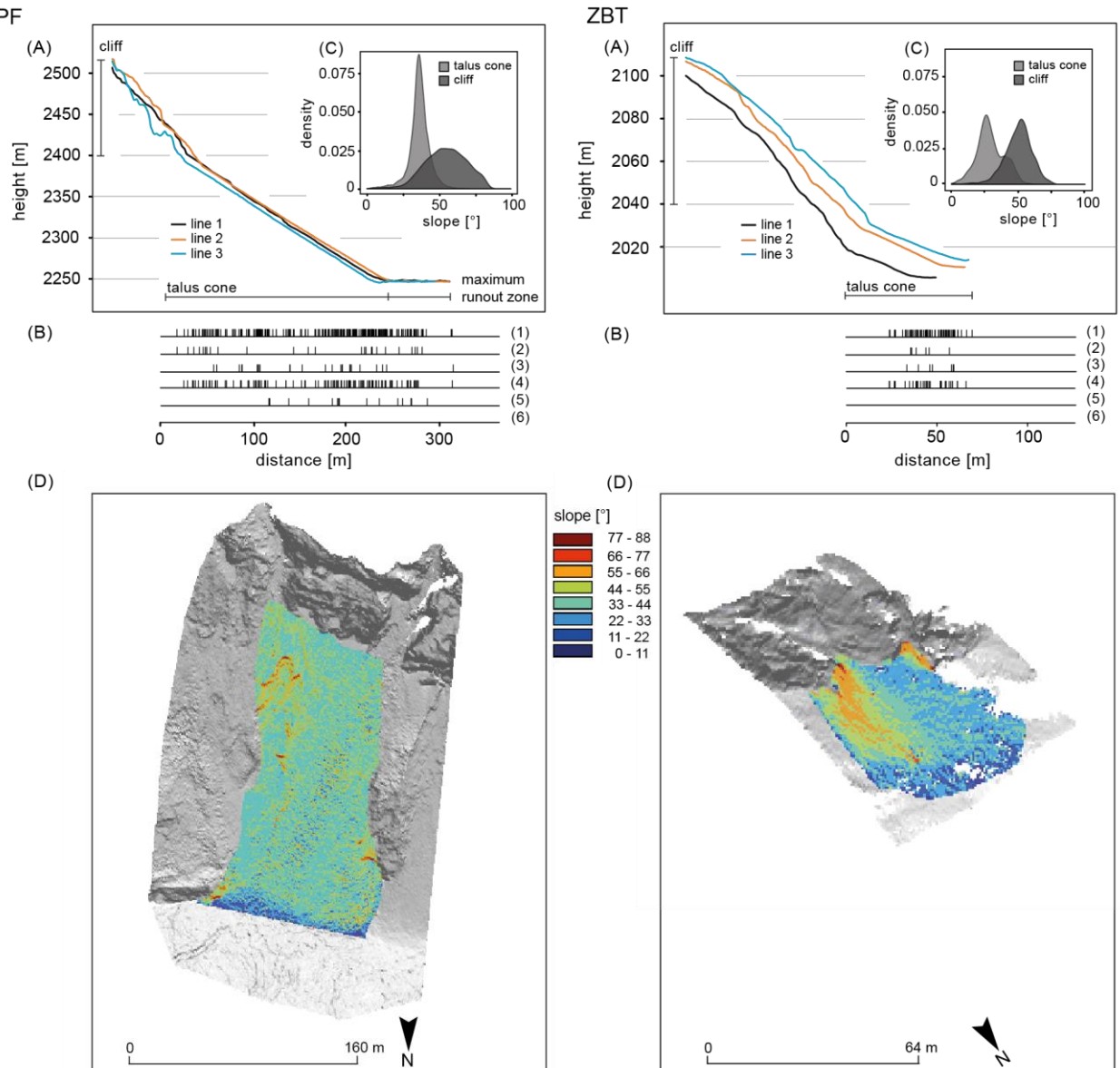

**Figure 5. Slope characteristics for the study sites PF and ZBT. Three swath longitudinal profiles from the cliff to the maximum runout distance (A). Statistical distribution of all measured blocks regarding their shapes and volumes on the talus cone. Every single line represents a measured block and thus the deposition on the slope can be illustrated. For this purpose, we have presented all blocks (1), the most (2) and least (3) spheroidal blocks and we have classified the block volume according to 10 m³ (4), 10² m³ (5) and 10³ m³ (6) (B). Kernel density estimation of slope inclination distinguished according to the cliff and the talus cone (C). Maps of slope inclination with different classes (D).**

### DTS and GA

The talus cones DTS and GA show quite similar values with 31.9° and 31.6°. Gerber (1974) measured 32° for talus cones consisting of limestone material, which corresponds to the study site of GA and DTS. Serrano et al. (2019) described in their

study slope inclination for limestone formations between 32°–36° and Knoblich (1975) gives values of 28°–43°. The two slopes are also similar regarding the longitudinal profiles (Fig. 6A). GA has a slightly basal concave talus cone downslope, while DTS is more straight.

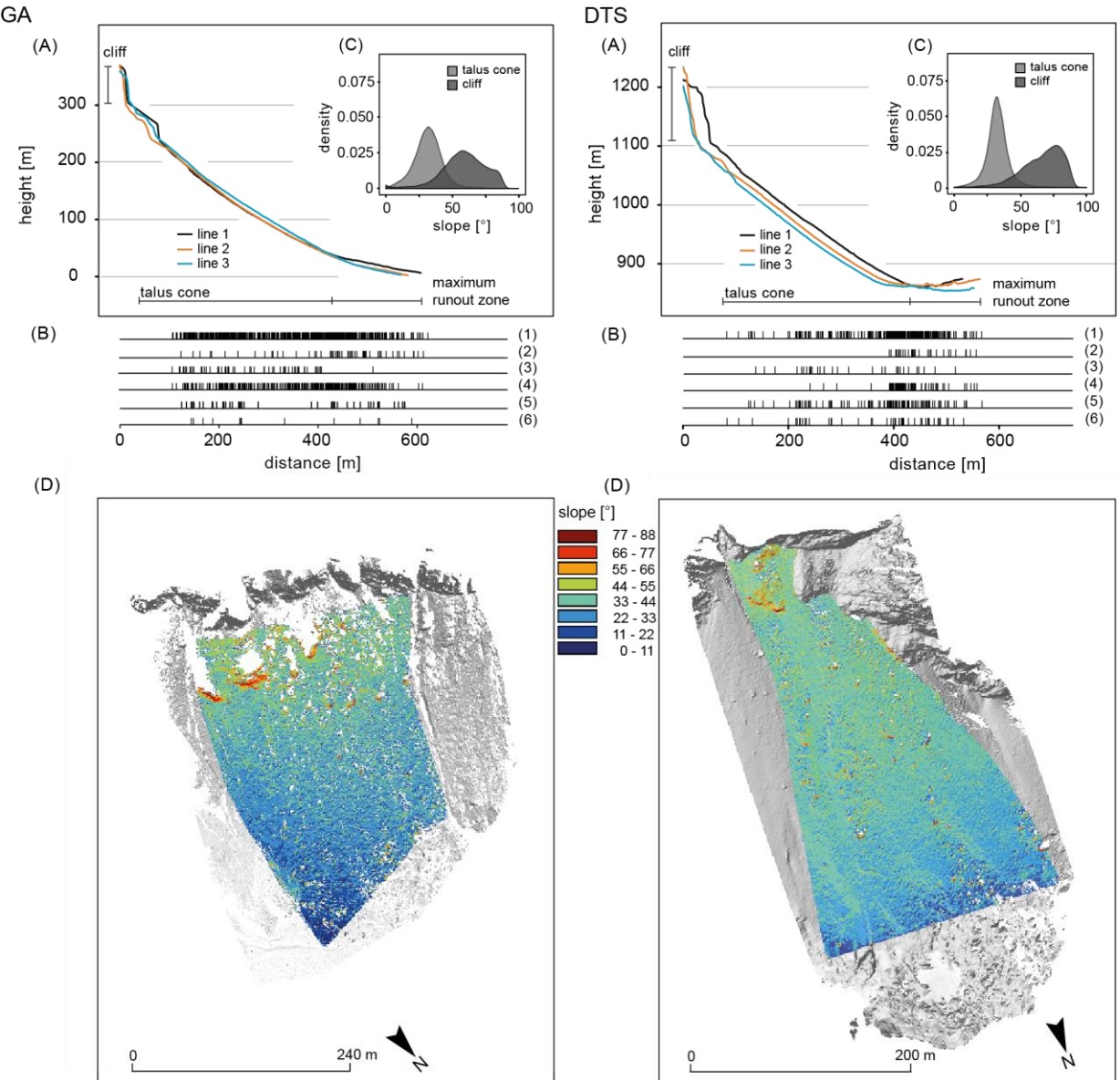

Figure 6. Slope characteristics for the study sites GA and DTS. Three swath longitudinal profiles from the cliff to the maximum runout distance (A). Statistical distribution of all measured blocks regarding their shapes and volumes on the talus cone. Every single line represents a measured block and thus the deposition on the slope can be illustrated. For this purpose, we have presented all blocks (1), the most (2) and least (3) spheroidal blocks and we have classified the block volume according to 10 m³ (4), 10² m³ (5) and 10³ m³ (6) (B). Kernel density estimation of slope inclination distinguished according to the cliff and the talus cone (C). Maps of slope inclination with different classes (D).

## 4.2 Relationship between block size, block shape and runout length

### 4.2.1 Analysis of individual boulders

In order to analyse the relationship between block volume, block shape and runout length, we calculated a Spearman rank correlation (Table 5). The results show only weak correlations between block volume and runout distance as well as weak correlations between runout length and block shape for the four test sites, which indicate no monocausal relationship.

**Table 5. Analysis of the relationship of block volume, axial ratio against runout length by means of Spearman rank correlation for each study site.**

|                        | PF   | ZBT      | GA       | DTS      |
|------------------------|------|----------|----------|----------|
| **Block volume rho**     | 0.15 | 0.33     | -0.04    | -0.29    |
| **Block volume p-Value** | 0.02 | 0.007111 | 0.39     | <<0.001  |
| **Axial ratio rho**      | 0.06 | 0.27     | -0.35    | -0.42    |
| **Axial ratio p-Value**  | 0.36 | 0.03     | <<0.001  | <<0.001  |

These results are in contrast to other studies, which state that longitudinal sorting of talus cones shows an increase of block sizes downslope (Jomelli and Francou (2000). Popescu et al. (2017) conclude that there is a gradual increase of boulder size towards the slope base, which is also stated by Copons et al. (2009), determining a dependency of rock volume and runout distance. Serrano et al. (2019) showed that the distal part of a slope is characterized by the accumulation of large blocks. In contrast to these studies Caine (1967) found a decrease of blocks sizes with distance downslope, but his results are statistically insignificant. Messenzehl and Dikau (2017) showed, that there is a distinct downslope increase in block size and sphericity, which indicates a combination of these block characteristics govern the runout length. These contradictory statements are supported by the work of Meißl (1998), who described in her analysis that the shape and size of the blocks mainly influence the width and height of the jumping parabolas, the rolling speed and the timing of the change between jumping and rolling. Meißl (1998) has also observed that larger blocks are not always reaching the longest runout distances. Possible causes could be that the blocks tend to sink, depending on the slope properties, but also interactions between the blocks cannot be excluded. Moreover, the fracturing of larger blocks can additionally lead to a loss of kinetic energy, which influences the disposition of blocks (Meißl, 1998).

In order to get a broader view on the data and in accordance to Haas et al. (2012) we plotted the relative distance against log10 block volume to show the relationship for every single talus cone in more detail (Fig. 7, 8). We combined this analysis with the axial ratio of the boulders. Figure 7A and Figure 8A show boxplots with six different quantile classes (<q10, q10–q25, q25–q50, q50–q75, q75–q90, >q90), which correspond to the 10, 25, 50, 75 und 90 % quantiles of the log10 block volume and log10 axial ratio. To analyze the relationship of the parameters in more detail, we highlighted the 10 % and the 90 % quantile

classes, which indicate blocks with a low (10%) and high (90%) axial ratio. These different classes are visualized with different symbols and colours in Figure 7B and Figure 8B.

Figure 7 and Figure 8 show that we can subdivide our study sites in two classes, regarding the runout length, block size and shape. The first class represent the study sites where the blocks can be reliably assigned to one bigger rockfall event (GA and DTS). The second class consists of the sites PF and ZBT, where we cannot reliably assign the boulders to one event and therefore have to assume continuous rockfall activity.

In the areas ZBT and PF the block sizes seem to have a recognizable influence on the runout length. Although the dispersion
of the data and the Spearman-Rank correlation show that this correlation is not significant (Table 5), large blocks above a certain size are not present in the short runout lengths. However, Figure 7B also shows that for PF, smaller and larger blocks are deposited in all areas of the slope. At the same time the block shape does not seem to play a major role for the runout distance, which is also indicated by the Spearman rank correlation, PF: r = 0.06, ZBT: r = 0.27. Cuboid formed and elongated blocks can actually be found over the entire talus cone without any visible clustering (Fig. 7B).

GA and DTS show that the size of the blocks obviously does not play a major role for the runout length on the talus cones, as the median values do not show a significant trend, but the dispersion of the runout distances clearly increases with increasing block size. This is visible in both areas, but is more pronounced at DTS. Beside this, it is also visible that blocks with larger volumes are also having smaller runout distances (Fig. 7B, 8B). The scatterplots show, that the block size scatters strongly over the entire talus cone. The block size can therefore not be used as an explanation for the runout distance alone, which is in
agreement with the Spearman rank correlation and supports our thesis above that there is no monocausal relationship.

The situation is different regarding the block shapes in the areas of GA and DTS. Here it is clearly visible that with increase in axial ratio, the runout distance decreases. This is also quite consistent with the results of the Spearman-Rank correlation, because in these two areas a slight correlation between axial ratio and range can be stated (Table 5). This is also visible in Figure 7B and 8B, which show the cuboid formed and longest blocks combined with the ranges: blocks with a low axial ratio
(<q10) reach larger distances than elongated blocks (>q90) (e.g. Pérez 1998). This can be seen in both areas from a relative distance of about 0.7 on the talus cone. We observed that the parameter axial ratio acts like a moderating parameter with regard to the deposition of blocks. Pérez (1998) and Messenzehl and Dikau (2017) conclude in their study that equant formed blocks are deposited predominantly at the talus toe. As already shown by Glover et al. (2015a) in their rockfall simulation, uniformly shaped blocks followed by elongated blocks achieve the largest runout distances. Platy blocks, on the other hand, are known
to tend to achieve long ranges only if they manage to place themselves on the shortest axis as the axis of rotation, which will probably not be the case for all blocks. This probably explains, why such blocks tend to have shorter runout distances (Glover et al., 2015a). It is particularly noticeable, however, that in the DTS area all blocks (<q10) with a low axial ratio achieve very high distances and are not found at all in the shorter distances (Fig. 8B). This supports the thesis above that play blocks only achieve high runout distances when they are placed on the short axis (wheel effect). This is different in the GA area. Here the
majority of the cuboid formed blocks can be found at the end of the slope, but also at the upper slope (Fig. 8B). This could be explained by the fact, that the blocks collide with each other during a large rockfall event, thereby dissipating their energy or,

maybe split into smaller blocks, resulting in the different block shapes being deposited in both the lower and the upper area of the talus cone (Ruiz-Carulla and Corominas, 2020). Although this cannot be definitively answered with the present study, the eyewitness reports speak in favour of the fact that such a collision of blocks did indeed take place at least at GA.


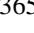

**Figure 7. Boxplots of relative distance versus log10 block volume and log10 block shapes of the measured rocks for the study sites of PF and ZBT (A). The red coloured points correspond to each measured individual block. The thick black lines within the boxes show the median bounded by the 1st quartile (25th percentile) and the 3rd quartile (75th percentile). The boxes show the interquartile range (IQR). The whiskers of the boxplots mark the minima and maxima of the data. The circles represent data that exceed the 1.5fold IQR (A). Scatterplot of relative distance versus log10 block volume for all study sites. Blocks with a low axial ratio (<q10) and a high axial ratio (>q90) are highlighted with different colours and symbols (B).**


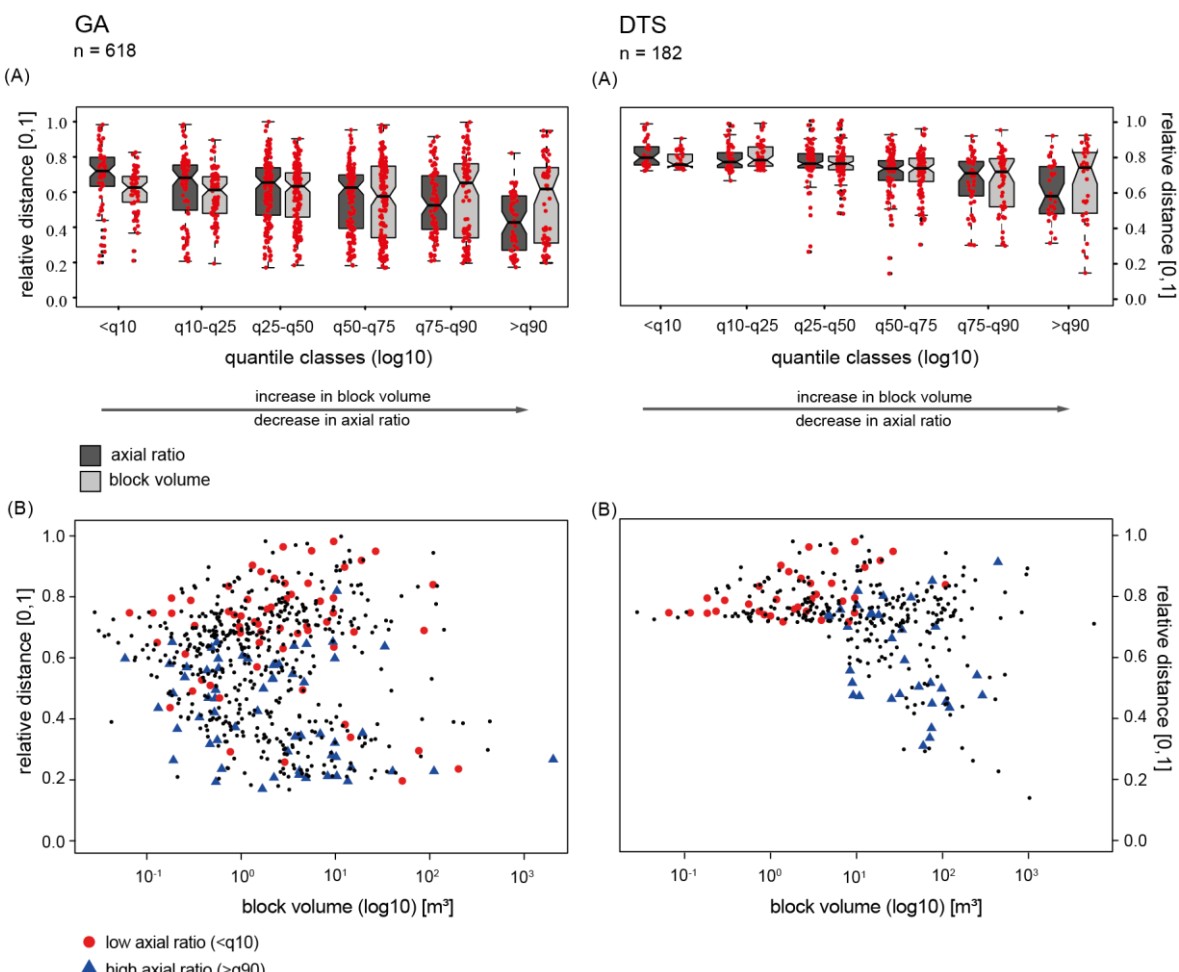

**Figure 8. Boxplots of relative distance versus log10 block volume and log10 block shapes of the measured rocks for the study sites of GA and DTS (A). The red coloured points correspond to each measured individual block. The thick black lines within the boxes show the median bounded by the 1st quartile (25th percentile) and the 3rd quartile (75th percentile). The boxes show the interquartile range (IQR). The whiskers of the boxplots mark the minima and maxima of the data. The circles represent data that exceed the 1.5fold IQR (A). Scatterplot of relative distance versus log10 block volume for all study sites. Blocks with a low axial ratio (<q10) and a high axial ratio (>q90) are highlighted with different colours and symbols (B).**

### 4.2.2 Roughness as indicator for block size distribution

To put the sampling analysis into a broader context and to verify if our single block analysis is representative for the entire talus cones, we attempted to use our high-resolution data to calculate roughness area-wide for all debris cones. Here, we understand roughness as an indicator of block size, with high roughness corresponding to large blocks and low roughness corresponding to smaller blocks.

The derived roughness based on the TLS point clouds shows a clear difference between the study sites (Fig. 9, 10), which we also found in our single block analysis in chapter 4.2.1. The highest roughness values on the talus cones GA and DTS can be found at the upper slope, somewhat less pronounced in ZBT, then fall off and rise again towards the end of the slope. This is

in good agreement with our single block analysis, where bigger blocks tend to have longer runout distances, but with some of the bigger blocks show shorter runout distances (Fig. 7, 8). In contrast to the other talus cones the roughness in PF and ZBT

seems to be continuously increasing in the downslope direction with the lowest roughness at the upper slope and the highest roughness at the end of the slope (Fig. 9A, 9B). This is also in good agreement with our results of the single block analysis in chapter 4.2.1, where we found a recognizable influence of the runout distance by the block volume (Fig. 7). It is also visible, that ZBT shows the lowest roughness values, which fits very well with the platy structure of the gneisses, since these blocks are mostly deposited with the shortest axis perpendicular on the talus cone. Wang and Lee (2010) and Mikoš et al. (2006)

simulate in their study that the roughness of the slope has an influence on the trajectory of the blocks and accordingly influences the deposition. In their laboratory experiment, Gratchev and Saeidi (2019) also come to the conclusion that the surface properties influence the rebound angle and, accordingly, the trajectory of rockfall material.

As our data show the talus cones only for one time step and we do not have information about the pre-event roughness, we cannot draw any conclusion how the roughness of the talus cone acts as a moderating factor for the runout distances of our

single blocks. But regarding Statham (1976) it is very likely that the roughness can play a major role and thus influenced the runout distance of blocks in the past and will do it also in the future. However, it must be taken into account that the influence of roughness is also likely to depend on the block size of the rockfall material. It must be considered, that the influence of roughness for smaller particles on the slope is more decisive than for larger particles that do not get stopped by coarser material (Statham, 1976).

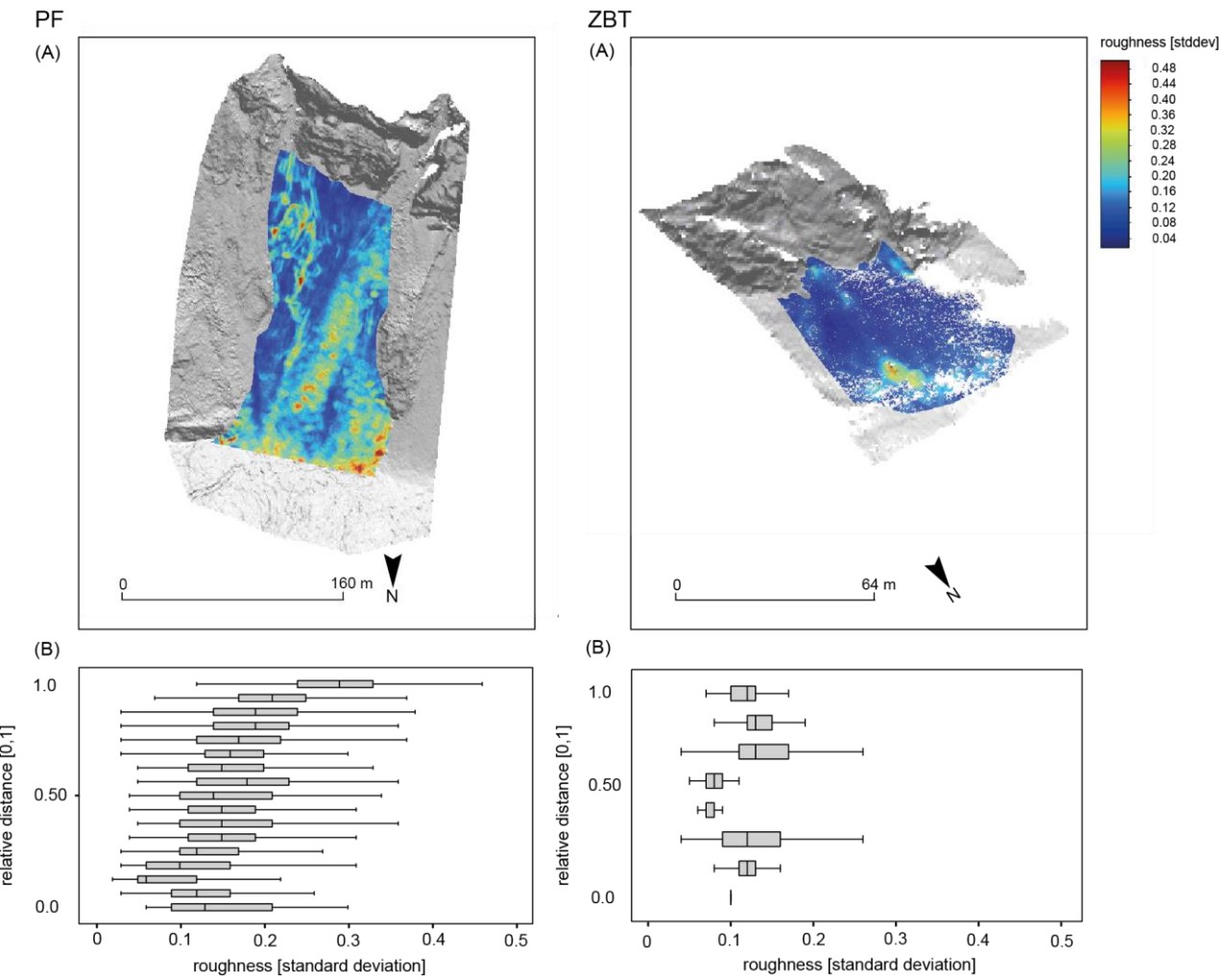


**Figure 9. Maps of classified slope inclination for the talus cones of PF and ZBT (A). Calculated roughness (standard deviation) based on the TLS point clouds which are classified to 10 m classes of the runout distance. To compare these values, we normalized the runout length to [0,1] (B).**

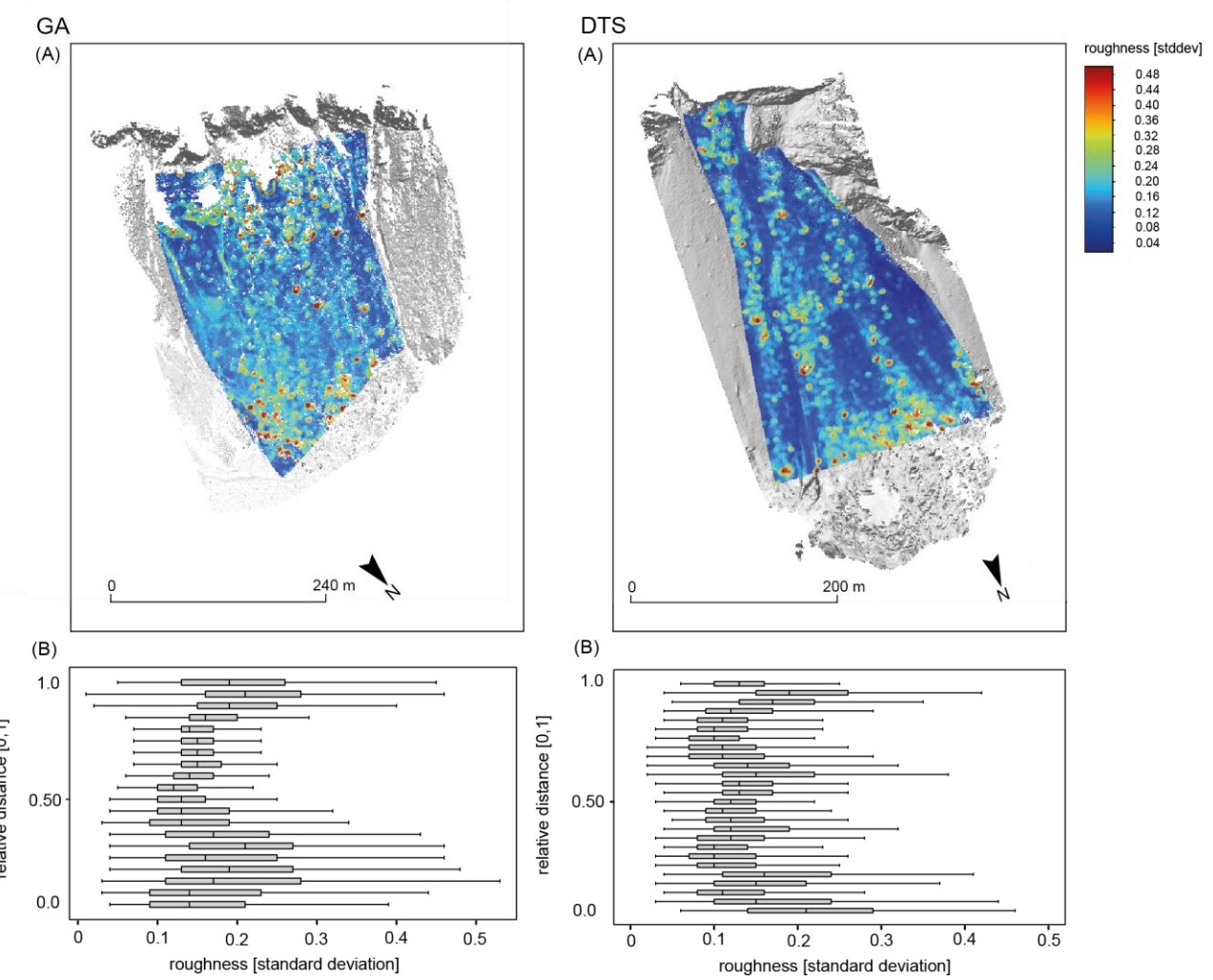

Figure 10. Maps of classified slope inclination for the talus cones of GA and DTS (A). Calculated roughness (standard deviation) based on the TLS point clouds which are classified to 10 m classes of the runout distance. To compare these values, we normalized the runout length to [0,1] (B).

## 5. Conclusion

Our investigations show a discernible relationship between the lithology and the characteristics of the rockfall material within the four test sites. We found larger blocks on the talus cones of GA and DTS with its thick banked limestones and dolomites. In contrast smaller blocks were found in areas with slated and platy gneisses (ZBT) as well as in thin layered basaltic lava material (PF).

In addition to block size, lithology also seems to have a significant effect on block shape. Thus, the highest axial ratios were found in the area of ZBT with platy gneisses. The limestones of DTS and GA show similar mean values and scatter of data,

indicating that dolomites and Wetterstein limestones produce similar block sizes and block shapes as thick-bedded reef limestones. Just the scatter of the data and the block size forces that the block volume and the block shape are not only controlled by the banking, which indeed tends to lead to large block sizes. An additional controlling variable seems to be the clefting of the material. The combination of irregular clefting and thick banks apparently result in blocks with large volumes but different axial ratios. In contrast to the other test sites, the material of PF with its small block sizes and low axis ratios

differs clearly. Here, the thin stratification of the lava material in connection with the strong tectonic stress due to frequent earthquakes and the resulting numerous fractures in the material certainly play a significant role for block size and block shape, resulting in smaller volumes and regular shaped boulders.

Our analyses reveal a complex relation between block size and block shape with respect to the runout distance of deposited blocks with divergences between different lithological settings. Compared to other studies (Whitehouse and McSaveney, 1983,

Jomelli and Francou, 2000, Sanders et al., 2009, Luckman, 2013a, Messenzehl and Dikau, 2017, Popescu et al., 2017) and with respect to our results we can neither confirm nor reject the theory of gravitational sorting (Fig. 7B, 8B) as we did not find a clear relationship between block size and runout distance.

But for PF, GA and DTS we can confirm a tendency toward gravitational sorting in a varying degree, since blocks with a higher block volume tend to have longer runout distances. But we also find larger blocks with shorter runout distance as well

as smaller blocks with longer runout distances. In contrast, almost no gravitational sorting could be detected for the ZBT area. Thus and in agreement with other studies (e.g. Meißl, 1998, Haas et al., 2012, Glover et al., 2015a, Messenzehl and Dikau, 2017) we can assume, that, the block size does not seem to be the only control variable.

Due to our findings, the axial ratio of boulders for GA and DTS seems to influence the deposition of rock fragments on the talus cone such that blocks with increasing axial ratio have decreasing runout distances (Pérez, 1998, Glover et al., 2015a,

Messenzehl and Dikau, 2017, Volkwein et al., 2018). It seems that boulders with a low axial ratio roll over any axis and do not lose momentum, while boulders with a high axial ratio seem to either not place themselves always on the favourable axis of rotation or loose energy by tumbling due to their unfavourable axial ratio. In our study, blocks of the study site DTS with a low axial ratio achieve large runout distances. For the study site of GA, blocks with a low axial ratio are found in the upper and in the lower part of the slope. But in case of PF and ZBT we cannot confirm the hypothesis of a moderating role of the

axial ratio. For both study sites cuboid formed blocks as well as elongated formed blocks are deposited over the entire slopes. Whether this is due to the different lithological setting or the fact that GA as well as DTS primarily include blocks of one rockfall event, while the blocks in PF and ZBT represent several individual and presumably temporally unrelated block falls, cannot finally be clarified. For this purpose, singular rockfalls in different lithologies should be investigated in further studies. The spatial roughness analysis based on the TLS point clouds show a good agreement with single block analysis regarding the

dependencies between runout distance and block volume. Thus we can conclude, that the roughness can be used for such analysis instead of time consuming single block measurements. By using roughness as a proxy for block size, it would therefore be possible to investigate a larger number of test areas with this method and thus to investigate the implied relationships between lithology and block size in more depth in future studies. Such studies could perhaps be supplemented by a sample

analysis of block shapes. Furthermore, due to the increasing number of available LiDAR data sets it seems possible to include
roughness as an additional influencing factor in a runout distance analysis, since information about the relief before an event would already be available for future rockfall events. We are quite sure, that such a study can be done on the base of ALS data or photogrammetric elevation models based on aerial photographs, where point densities are high enough to resolve blocks of a certain size, which are nowadays available for the whole Alps and mountain ranges worldwide.

**Data availability.**

The data used in this study are accessible upon request by contacting Kerstin Wegner (KWegner@ku.de).

**Author contributions.**

KW, FH and TH designed the conceptual idea of the manuscript. KW and FH collected the terrestrial laser scanning data on La Réunion. FH collected the TLS data of DTS, ZBT and GA. TH contributed to statistical analysis of data. VD, NV and PK acquired the dGPS data on La Réunion. NV, PK and AP contributed to fieldwork on La Réunion. AM contributed to the
financial support of the travel costs to La Réunion and gave feedback to the analysis of data, references, results and interpretation. MB contributed to the acquisition of the terrestrial laser scanner. KW wrote the manuscript with discussions and improvements from all co-authors.

**Competing interests.**

The authors declare that they have no conflict of interest.

**Acknowledgements.**

F. Haas kindly provided the photographs used for Figure 1, in particular of La Réunion, Dreitorspitze and Zwieselbach valley. The photo of Gampenalm was taken by an installed camera as part of the ERC SLIDEQUAKES project. The field work on La Réunion was funded by ERC-CG-2013-PE10-617472SLIDEQUAKES. This is IPGP contribution number 4204. The Open Access publication of this article was supported by the Open Access Fund of the Catholic University Eichstätt-Ingolstadt.

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
