# Peer review of "Assessing the effect of lithological setting, block characteristic and slope topography on the runout length of rockfalls in the Alps and on the La Réunion island"

_Natural Hazards and Earth System Sciences, 2020_

## Referee Comment (RC1) · Anonymous Referee #1 · 27 Oct 2020

This paper presents a detailed analysis of deposited blocks on four study sites that are very different in terms of lithology and topography. The objective was to study the relationships between block propagation distances on the one hand and block properties (size and shape indicators) and topography indicators (mean slope, slope distribution, roughness, in particular).

The authors provided very detailed measurements of the above mentioned quantities based on analyses of Lidar data. The results obtained are very interesting as they differ from previous studies and thus moderate these results. The authors discuss

these interesting results in details.

I recommend publication of the paper provided that the following minor comments are integrated :

1) The calculation of topographical indicators (sections 3.2 and 3.3) are not detailed enough in my opinion. The authors only refer to previous studies. Additional information could be provided.

2) The figures are small and difficult to read (small font size, in particular - Figure 4,5, 6 and 7, in particular). They have to be improved.

3) Because the relationships between the run-out distances and topography/block properties are not clear, section 4.2 is very difficult to follow. This section should be clarified.

4) The authors mention that the deposited blocks are not only due to single blocks propagations, despite the analyses are based on this assumption. This point should be more emphasized / discussed.

5) p.5 l.101 : "show indicate " → typo ?

---

## Referee Comment (RC2) · Anonymous Referee #2 · 27 Oct 2020

The presented manuscript deals with the four analysed talus slopes and a thorough mapping of lithological characteristics, being deposited rock shapes and sizes and its influence on runout length. The manuscript presents itself thoroughly written and elaborated. The presented data is of very good quality and meticulously analysed.

The findings corroborate a complex interaction between analysed parameters such as rock shape, sphericity, slope characteristics, above all surface roughness. The authors aim to contrast the findings in previous work, which is highly favourable. As the findings are complex to trim into existing results, they fail to a certain extent to present the new

aspects in a consistent way.

Main concerns: 1) Rockfall source and site consistency: The reasoning for the selection of the presented sites was not argued for. The impression, that those were randomly chosen sites due to logistic reason rather than careful scientific screeing is imminent. The authors state, that "For the two study sites PF and ZBT the blocks cannot be assigned to one single rockfall. Whereas the blocks of the other two study sites GA and DTS can be assigned to a rockfall event". This is a key difference between the sites as the processes are significantly different. Single block fall as opposed to block fall with fragmentation as opposed to rock avalanches – here we would need to differentiate within the volume classes again – are governed by different kinematic behaviour. The authors are strongly urged to focus on these differences between their sites and comparing it to respective previous literature. Was gravitational sorting only seen on "single rockfall" talus slopes? Is the data quality good enough to argue for or against it?

2) Key finding: Although it is highly appreciated to publish work not in line with previous findings, the discrepancies, differences, etc. are to be highlighted in a more consistent manner. Purely publishing a scientific "it's complicated" is insufficient. The impression is that they applied plotting schemes and analysis methodologies found in previous literature in order to make comparison easier. This is certainly done with good intentions, but the meandering presentation of the results is suboptimal.

3) Link to surface roughness: The authors posess excessive data sets and high resolution DEM in particular. The publication quality would gain significance and make use of the available data set if sections 4.1 and 4.2 would be linken in a meaningful way. The link between surface roughness and block size/shape and runout length is a key factor in talus slopes. The naïve understanding of a rockfall propagating on a talus slope is, that as long as the roughness dimension, i.e. lying boulders, etc. is smaller than the travelling block axis length, the breaking effect is rather small. This data set would provide a perfect investigation basis for those dependencies. Runout length

vs. surface roughness, coupled with different shape classes, different masses. Rather then presenting a plain characteristicss of the talus slopes as in Fig. 5, the extracted correlations - if there are any – would be of interest for the NHESS community. In this line, the presentation of the analyses, in particular Figure 5 and 7 should be revised.

Generally, the presented manuscript might merit publication if the main concerns are addressed and presented in a consistent analysis. Contrasting results are desirable as they lead to a discussion in the community but a sound reasoning is a pre-requisite.

In the following there are some minor/technical/content corrections with might become obsolete after major revisions:

General comment on the use of parentheses: Clearly a matter of writing style, however, IMHO the excessive use of parentheses hinders the reading flow. Personal guidance is: if it's important, rephrase it into the written sentences, if it does not merit being included in the text, remove it. The authors might check their use of parentheses with this in mind, or discard it as the referee's spleen. Does not hold for introduction of acronyms, of course.

Figure font sizes: Revise the font size and general sizing of heavily loaded figures.

Abstract: l15: is explanation of LiDAR in abstract necessary? l19: no parantheses – if necessary, add it to text ". . . and longitudinal profile curves" l19-20: Start concluding sentences with what could be confirmed. l23,23: no acronyms in abstract.

1. Introduction L30: can be deposited on storage landforms such as talus slopes. If playing the devil's advocate one could argue that all rockfalls, which have space to be deposited on a storage landform are from a natural hazard's perspective irrelevant, as they do not threaten any infrastructure. l31ff Although citing relevant articles is a form of contemplating on previous work and arranging the presented work in the current state of the art research, citing it after a common senses statement like " a falling rock is dangerous" should be omitted. Rather cite something specific, if at all necessary to

include in such a general introduction. l39 probably "global climate change", but is it really? Isn't just the development pressure by increasing land use and a diminishing risk tolerance in societies in general the leading factors why natural hazards have to be mapped ever more accurately? And of course because the technical means improve constantly... L53 "conditions of the talus cone" → soil characteristics in general. The presented study focuses on runout on talus cones, but rockfall runout happens on more terrains than talus. L55: runout trajectory is predominantly governed by the impact conditions, geometrical boundary conditions and soil-rock interaction. Thus, the moment of inertia plays a role on the specific kinematical properties – meaning how fast can a rock spin, how reactive is a rock to slope ruggedness, etc - but ultimately does not alter significantly as a single leading factor the runout trajectory. L70 introduce the test sites with names and acronyms here.

2. Study Sites l85 Figure1: Are there aerial maps available for all sites? Or photographs in the same aspect ratio? The acessebilty of the image would improve if consistency with respect to aspect ratio, presentation style (arial with indicated releasa and depsostion area – or 1 photograph per site with a similar viewing angle. l90 Figure 2: Is a consistent presentation in one figure possible? L100 delete "show" or " indicate" Table 1: totally aware, that formatting tables is a nightmare, but "centered" titles columns would improve readability. L135 x,y, z are variables – use italic font L137 z is a variabl, use italic font L139 points/m2 → pts/m2 as used in the table. Table 3: spaces in between the scan resolution values, referencing precision, use \cdot or similar for Number of points raw data set or 1.5e6 notation.

3. Determination of block size, block shape and runout length l144: a,b and c are parameters use italic font l146 every block with c > 0.5 m? Shortest axis large than 0.5 m? l150 Figure 3: I basically see a cuboid. If you really want to show how the bounding box of a given rock is defined, show a bounding box with labels around a rock. Wasting a figure to print a cuboid is a bit questionable. The NHESS reader should be able to imagine a bounding box around a rock though. L163: normalization of runout: is

the interval [0,1] defined as minimal, maximal runout or is it 0 = no runout meaning stopping in release area (which probably is not possible) and 1 is maximal measured runout?

4. Results and Discussionl L186 longest axis greater than 0.5 m? In the plot c is the shortest axis.. Clarify? A criteria on the resolution boundary, i.e. the smallest axis would make more sense. l217: Add labelling (1),(2), ..., (6) in all subpanels. Figure 6: Are the depicted areas only the talus fan? Could some surrounding be included and the talus extent be marked with a dashed line equivalently as in Fig. 1? Colour code and scale range make the talus slope look quite uniform. Could the range be shortened and thus differences in the main talus deposition become more obvious? What is the slope inclination grid size base raster? Could the indications of steepness classes help the reader to classify the slopes visually more easily? L297ff Rockfall source apparently are two different processes: single block fall and rock avalanches. This fact changes the entire approach. Are differences due to this fact?

5. Conclusion L357 boulders with low axial ratios do also have a predominant rotation axis, as a perfect symmetric rock does not exist in nature. It rather has no large flat areas, hindering a re-acceleration after a landing on such a flat surface.

---

## Referee Comment (RC3) · Anonymous Referee #3 · 27 Oct 2020

The paper presents detailed analysis of rockfall deposits mapped using point cloud datasets captured at four sites with differing lithology. The analysis explores how the properties of rock shape and volume along with terrain roughness and morphology influence runout distance. The findings are discussed and contrasted with previous works on the topic. Differences are identified in the runout character according to lithological setting. A number of analysis methods fail to link the measured metrics with the potential rockfall runout process controlling them and perhaps could be enhanced by applying alternative metrics of rock shape and volume. Inconsistencies in terms

applied to descriptions of rock shape lead to some confusions in understanding the findings regarding the influence of rock volume and shape on rockfall runout.

The following observations and questions arise: 1) The differences in lithology between the study sites make this study of interest for rockfall. However, the fragmentation of rock during rockfall and the lithological control on preconditioning down slope fragmentation is poorly treated in this work. It would be interesting to consider the rock-mass properties of each of the rock walls. How do the site specific failure mechanisms, discontinuities and rock strength influence the distribution of rock fragments across the talus cone? Investigating this theme would be possible with the detail of data presented in this work, and have the potential to enhance the observations made in sections 4.1 and 4.2. 2) The use of rock axis ratios as a shape classification is an obvious choice given the volume of available data. Moreover, the geometric axis ratio has a bearing on the inertial axes of the rocks and its ability to maintain momentum during run out. The rock volume and shape are therefore coupled in their influence on runout behaviour. The analysis could benefit from coupling rock volume and shape. General comments to the text. 1. Introduction L34 to L38 The text introduces preconditioning and preparatory factors as well as triggering events. A clear distinction between preconditioning and triggering rockfalls for the examples of each that are given would be useful. L39 "Due . . . importance", of what? L61 -63 please clarify the use of proximal in this sentence, proximal to the source or deposits?

2. Study sites L97 & 98 Please clarify the use of "untreated" with regards the rock mass description. Table 1 could benefit from details of the rock mass properties and strength.

3. Materials and Methods L122 consider replacing "Both. . ." with "Each scanner works. . .". L124 clarity on the importance of coloured point clouds to the methodology would be useful. Section 3.2 the method applied to obtain rock volume using the three principal geometric axes a b & c give the volume of a cuboid. To what extent does this method over estimate rock volume in this study? L152 The applied rock

shape indicator bundles the platy and elongate rock shapes into the same indicator, please discuss this choice in more detail. L154 The axis ratio of 1 in which axes are the same length is referred to as a round shape. How does the applied shape classification account for the roundness of the rock? Could an axis ratio of 1 be described as "equant"? Please consider applying equant throughout the text. How do larger rocks with flat sides but axis ratio of 1 runout in comparison to a rounded rock? Section 3.3 L168 "rock fall" please replace with rockfall.

4. Results and Discussion L190 please consider deleting "dispersion also". L193 please consider "clearly shown" in replacement of "well visible". L241 To what extent can this be attributed to fragmentation of rock during runout? L300 repetitive use of adverb "clearly" "obviously".

5. Conclusions L345 the term sphericity is newly introduced in the conclusions, it is not clear how sphericity was measured in the methods. Is rounded or equant shape meant?

---

## Author Comment (AC1) · 18 Dec 2020

**Response to reviewer comments on "Assessing the effect of lithological setting, block characteristic and slope topography on the runout length of rockfalls in the Alps and on the La Réunion island"**

**Kerstin Wegner, Florian Haas, Tobias Heckmann, Anne Mangeney, Virginie Durand, Nicolas Villeneuve, Philippe Kowalski, Aline Peltier, Michael Becht**

nhess-2020-322

**Response to Anonymous Referee #1 (RC1)**

We thank the anonymous referee #1 (RC1) for the constructive comments and detailed feedback. We greatly appreciate the suggested revisions which will improve the quality of our revised manuscript. In the following the original comments of the reviewer #1 are listed point by point in blue and replied to in black.

This paper presents a detailed analysis of deposited blocks on four study sites that are very different in terms of lithology and topography. The objective was to study the relationships between block propagation distances on the one hand and block properties (size and shape indicators) and topography indicators (mean slope, slope distribution, roughness, in particular).

The authors provided very detailed measurements of the above mentioned quantities based on analyses of Lidar data. The results obtained are very interesting as they differ from previous studies and thus moderate these results. The authors discuss these interesting results in details.

I recommend publication of the paper provided that the following minor comments are integrated :

Review Comment 1:

1) The calculation of topographical indicators (sections 3.2 and 3.3) are not detailed enough in my opinion. The authors only refer to previous studies. Additional information could be provided.

Reply: We will provide additional information on the described parameters in sections 3.2 and 3.3.

Review Comment 2:

2) The figures are small and difficult to read (small font size, in particular - Figure 4,5, 6 and 7, in particular). They have to be improved.

Reply: We will revise the figures when we revise the manuscript and ensure good readability in this step.

Review Comment 3:

3) Because the relationships between the run-out distances and topography/block properties are not clear, section 4.2 is very difficult to follow. This section should be clarified.

Reply: By means of our analyses and figures we described the complex relationship between runout lengths, block characteristics and topographic conditions of the slope and compare this with the results of other studies. In the revised manuscript we will re-organise this section in a more structured way and clarify the section 4.2.

Review Comment 4:

4) The authors mention that the deposited blocks are not only due to single blocks propagations, despite the analyses are based on this assumption. This point should be more emphasized / discussed.

Reply: We fully acknowledge that our study sites differ with respect to rockfall processes: While at Gampenalm and Dreitorspitze, we clearly see deposits that were formed by one event, the blocks measured at the other two study sites were likely deposited onto a talus slope that has been formed by "continuous" rockfall activity during a longer period of time. In the La Réunion area, large rockfalls are triggered very often by the seismic conditions there, but these are not recorded in detail. In the Zwieselbach valley, it is not possible to assign the blocks to one event. Accordingly, we will discuss this point in more detail and we will pay particular attention to this distinction in the revision of the manuscript and revise corresponding sections.

Review Comment 5:

5) p.5 l.101 : "show indicate " ! typo ?

Rely: We will delete "show".

---

## Author Comment (AC2) · 18 Dec 2020

**Response to reviewer comments on "Assessing the effect of lithological setting, block characteristic and slope topography on the runout length of rockfalls in the Alps and on the La Réunion island"**

**Kerstin Wegner, Florian Haas, Tobias Heckmann, Anne Mangeney, Virginie Durand, Nicolas Villeneuve, Philippe Kowalski, Aline Peltier, Michael Becht**

**nhess-2020-322**

**Response to Anonymous Referee #2 (RC2)**

We thank the anonymous referee #2 (RC2) for the constructive comments and detailed feedback. We greatly appreciate the suggested revisions which will improve the quality of our revised manuscript. In the following the original comments of the reviewer #2 are listed point-by-point in blue and replied to in black.

The presented manuscript deals with the four analysed talus slopes and a thorough mapping of lithological characteristics, being deposited rock shapes and sizes and its influence on runout length. The manuscript presents itself thoroughly written and elaborated. The presented data is of very good quality and meticulously analysed. The findings corroborate a complex interaction between analysed parameters such as rock shape, sphericity, slope characteristics, above all surface roughness. The authors aim to contrast the findings in previous work, which is highly favourable. As the findings are complex to trim into existing results, they fail to a certain extent to present the new aspects in a consistent way.

**Main concerns**

Review Comment 1:

1) Rockfall source and site consistency: The reasoning for the selection of the presented sites was not argued for. The impression, that those were randomly chosen sites due to logistic reason rather than careful scientific screeing is imminent. The authors state, that "For the two study sites PF and ZBT the blocks cannot be assigned to one single rockfall. Whereas the blocks of the other two study sites GA and DTS can be assigned to a rockfall event". This is a key difference between the sites as the processes are significantly different. Single block fall as opposed to block fall with fragmentation as opposed to rock avalanches – here we would need to differentiate within the volume classes again – are governed by different kinematic behaviour. The authors are strongly urged to focus on these differences between their sites and comparing it to respective previous literature. Was gravitational sorting only seen on "single rockfall" talus slopes? Is the data quality good enough to argue for or against it?

Reply: We take the reviewer's comment very seriously. The study areas have in fact not been selected systematically. Rather than deliberately selecting areas with lithological or climatic differences in mind, we are pursuing a synthesis of various existing data with a very high quality. Even though we have study areas with different process activity, we do not want to separate the manuscript into two parts. One of our main goals was to focus on the different lithological conditions of the study sites, which are conspicuous. On the other hand, we fully acknowledge that the sites represent different processes such as rock topples (Gampenalm) and block fall. Rock avalanches (the other end member named by the reviewer), however, have definitely not taken place at our study sites. The particles that we analysed either represent single block falls or belong to a single event where a larger

rock mass detached from the rock face and probably further disintegrated during descent. As this point of criticism was also noted by the other reviewers, we intend to work out this difference in a more focused and structured way when revising the manuscript. We do think that we can make statements about gravitational sorting because of the high quality of the data. Moreover, we also found that parameters other than the block size influence the runout distances of the blocks.

Review Comment 2:

2) Key finding: Although it is highly appreciated to publish work not in line with previous findings, the discrepancies, differences, etc. are to be highlighted in a more consistent manner. Purely publishing a scientific "it's complicated" is insufficient. The impression is that they applied plotting schemes and analysis methodologies found in previous literature in order to make comparison easier. This is certainly done with good intentions, but the meandering presentation of the results is suboptimal.

Reply: For the revision, we will aim to organize the structure of the manuscript more stringently. This will be done especially in relation to the different process types that occur at the study sites. Accordingly, we will revise and restructure the introduction, the current state of research, the description of the study sites, the results and figures and the conclusion. Following the suggestion of the reviewer in the subsequent comment, we will better and more rigorously consider roughness as an additional factor that "complicates" particle runout.

Review Comment 3:

3) Link to surface roughness: The authors posess excessive data sets and high resolution DEM in particular. The publication quality would gain significance and make use of the available data set if sections 4.1 and 4.2 would be linken in a meaningful way. The link between surface roughness and block size/shape and runout length is a key factor in talus slopes. The naïve understanding of a rockfall propagating on a talus slope is, that as long as the roughness dimension, i.e. lying boulders, etc. is smaller than the travelling block axis length, the breaking effect is rather small. This data set would provide a perfect investigation basis for those dependencies. Runout length vs. surface roughness, coupled with different shape classes, different masses. Rather then presenting a plain characteristicss of the talus slopes as in Fig. 5, the extracted correlations - if there are any – would be of interest for the NHESS community. In this line, the presentation of the analyses, in particular Figure 5 and 7 should be revised.

Reply: Thank you for this comment and the advice. We have planned the following additional and more rigorous analysis for the revision of the manuscript in order to be able to assess the influence and the dependence of the roughness of the slope on the deposited blocks: We will get the most out of the original high-resolution TLS point cloud data to quantify surface roughness along the hillslope profile (from the rockwall down to the footslope of the talus) with two points in mind: First, roughness is expected to reflect the granulometry of the substrate at grain sizes smaller than the single blocks we measured. This parameter could yield more information regarding gravitational sorting patterns where it indicates a coarsening towards the footslope and/or a concentration of coarse material at the foot of the rockwall. We want to do this by using roughness as a proxy for grain size. This is to determine and analyze the sorting of particle sizes beyond the measured block sizes for the single debris cones. Second, especially for small blocks, their size relative to the surface roughness could be an additional factor influencing their runout length. For this purpose, the roughness along the calculated runout paths of each individual

block are to be determined in order to be able to analyze the influence of roughness on runout distance and how this interacts with the different block sizes. Accordingly, Figures 5 and 7 will be revised or replaced by others.

Generally, the presented manuscript might merit publication if the main concerns are addressed and presented in a consistent analysis. Contrasting results are desirable as they lead to a discussion in the community but a sound reasoning is a pre-requisite.

In the following there are some minor/technical/content corrections with might become obsolete after major revisions:

Review Comment 4:

General comment on the use of parentheses: Clearly a matter of writing style, however, IMHO the excessive use of parentheses hinders the reading flow. Personal guidance is: if it's important, rephrase it into the written sentences, if it does not merit being included in the text, remove it. The authors might check their use of parentheses with this in mind, or discard it as the referee's spleen. Does not hold for introduction of acronyms, of course.

Reply: We agree with the reviewer and we will review the manuscript and check the relevant sentences. In the according sentences where parentheses can be omitted, we will delete them or rephrase a new sentence.

Review Comment 5:

Figure font sizes: Revise the font size and general sizing of heavily loaded figures.

Reply: We will revise the figures.

**Abstract**

Review Comment 6 – l15:

is explanation of LiDAR in abstract necessary?

Reply: As LiDAR is a common technical term and as this is also specified in the NHESS English guidelines and house standards, we follow the reviewer's comment, delete the explanation and refer to it in the text at the first naming.

Review Comment 7 – l19:

no parantheses – if necessary, add it to text ". . .  and longitudinal profile curves"

Reply: We will rephrase this sentence.

Review Comment 8 – l19-20:

Start concluding sentences with what could be confirmed.

Reply: We will rephrase the sentence so that we start it with our determination.

Review Comment 9 – l23,23:

no acronyms in abstract.

Reply: We will delete the acronyms in the abstract and explain it at the first occurrence in the text.

**1. Introduction**

Review Comment 10 – l30:

*can be deposited on storage landforms such as talus slopes. If playing the devil's advocate one could argue that all rockfalls, which have space to be deposited on a storage landform are from a natural hazard's perspective irrelevant, as they do not threaten any infrastructure.*

Reply: We agree that not every rockfall event endangers human lives and infrastructure facilities. But nevertheless, research in areas where no infrastructure is involved also plays an important role in terms of the transferability of results. Moreover, the runout and deposition of rockfall is also a geomorphological interesting problem. We will rephrase the sentences and put them into perspective.

Review Comment 11 – l31ff:

*Although citing relevant articles is a form of contemplating on previous work and arranging the presented work in the current state of the art research, citing it after a common senses statement like " a falling rock is dangerous" should be omitted. Rather cite something specific, if at all necessary to include in such a general introduction.*

Reply: In the revised manuscript we will rephrase the addressed text passages.

Review Comment 12 – l39:

*probably "global climate change", but is it really? Isn't just the development pressure by increasing land use and a diminishing risk tolerance in societies in general the leading factors why natural hazards have to be mapped ever more accurately? And of course because the technical means improve constantly. . .*

Reply: We agree that spatial densification and corresponding infrastructure increase the problem of dealing with natural hazards. When we revise the introduction, we will structure it more in line with our analysis.

Review Comment 13 – l53:

*"conditions of the talus cone" → soil characteristics in general. The presented study focuses on runout on talus cones, but rockfall runout happens on more terrains than talus.*

Reply: We agree that rockfall runout can take place on other storage landforms than talus cones as well. We will rephrase this sentence and write "topographic conditions".

Review Comment 14 – l55:

*runout trajectory is predominantly governed by the impact conditions, geometrical boundary conditions and soil-rock interaction. Thus, the moment of inertia plays a role on the specific kinematical properties – meaning how fast can a rock spin, how reactive is a rock to slope ruggedness, etc - but ultimately does not alter significantly as a single leading factor the runout trajectory.*

Reply: We will check the parameters mentioned in the literature and will revise and improve the sentences according to the section. We will describe the influencing parameters regarding the runout length of blocks in more detail.

Review Comment 15 – l70:

*Introduce the test sites with names and acronyms here.*

Reply: We will follow the reviewer's suggestion.

**2. Study Sites**

Review Comment 16 – l85:

Figure1: Are there aerial maps available for all sites? Or photographs in the same aspect ratio? The acessebilty of the image would improve if consistency with respect to aspect ratio, presentation style (arial with indicated releasa and depsostion area – or 1 photograph per site with a similar viewing angle.

Reply: We will try to create consistent figures of the study areas.

Review Comment 17 – l90:

Figure 2: Is a consistent presentation in one figure possible?

Reply: We will change the figures and present all four study sites in one common figure.

Review Comment 18 – l100:

delete "show" or " indicate"

Reply: We will delete "show".

Review Comment 19:

Table 1: totally aware, that formatting tables is a nightmare, but "centered" titles columns would improve readability.

Reply: We will centre the column headings in the revised manuscript.

Review Comment 20 – l135:

x,y, z are variables – use italic font

Reply: We will follow the reviewer's comment and write the variables in italic font.

Review Comment 21 – l137:

z is a variabl, use italic font

Reply: We will follow the reviewer's comment and write the variable in italic font.

Review Comment 22 – l139:

points/m2 → pts/m2 as used in the table.

Reply: We will follow the reviewer's comment and write "pts/m$^2$" as shown in Table 3.

Review Comment 23:

Table 3: spaces in between the scan resolution values, referencing precision, use \cdot or similar for Number of points raw data set or 1.5e6 notation.

Reply: We will follow the comment. We will insert spaces in between the scan resolution and referencing precision values and use a uniform notation for the number of points raw data set.

**3. Determination of block size, block shape and runout length**

Review Comment 24 – l144:

a,b and c are parameters use italic font

Reply: We will follow the comment and write the variables in italic font.

Review Comment 25 – l146:

every block with c > 0.5 m? Shortest axis large than 0.5 m?

Reply: The "c." does not refer to the c axis but is the abbreviation of "circa". We have measured those blocks in the coloured point cloud that have a longest axis longer than ~0.5 m. We will remove the parentheses here and rephrase the sentence.

Review Comment 26 – l150:

Figure 3: I basically see a cuboid. If you really want to show how the bounding box of a given rock is defined, show a bounding box with labels around a rock. Wasting a figure to print a cuboid is a bit questionable. The NHESS reader should be able to imagine a bounding box around a rock though.

Reply: We will revise Figure 3. We will represent the axes either by a block from the point cloud or by a photograph. We are considering whether this representation could be integrated into the rearranged Figures of the study sites.

Review Comment 27 – l163:

normalization of runout: is the interval [0,1] defined as minimal, maximal runout or is it 0 = no runout meaning stopping in release area (which probably is not possible) and 1 is maximal measured runout?

Reply: Our purpose was to enable the comparability of the runout length of all study sites since they differ greatly in length. For this reason, we have normalized them. We have assumed that "0" stands for the detachment area or the area between the rock face and the slope and "1" for the maximum runout length of a block. Accordingly, no block is assigned the value "0". We will describe this point in the revised manuscript in more detail.

**4. Results and Discussion**

Review Comment 28 – l186:

In the plot c is the shortest axis.. Clarify? A criteria on the resolution boundary, i.e. the smallest axis would make more sense.

Reply: We have selected blocks whose longest axis measures at least 0.5 m. Please see our response to comment 25 l146.

Review Comment 29 – l217:

Add labelling (1),(2), : : :, (6) in all subpanels.

Reply: In the revised manuscript we will provide subpanels 1-6 in Figure 5 for each study area for the statistical distribution of the blocks regarding their block volumes and shapes.

Review Comment 30:

Figure 6: Are the depicted areas only the talus fan? Could some surrounding be included and the talus extent be marked with a dashed line equivalently as in Fig. 1? Colour code and scale range make the talus slope look quite uniform. Could the range be shortened and thus differences in the main talus deposition become more obvious?

What is the slope inclination grid size base raster? Could the indications of steepness classes help the reader to classify the slopes visually more easily?

Reply: The slopes shown in Figure 6, represents only the area up to the talus toe. For the revision, we propose to add an analytical hillshade behind the maps and extend the area compared to the Figures 1 and 2. We derived the slope based on a grid with a resolution of 0.75 m. We agree with the reviewer that classes for the slope maps will improve the visualization to better show the differences between each slope. We will implement this in the revision.

Review Comment 31 – l297ff:

Rockfall source apparently are two different processes: single block fall and rock avalanches. This fact changes the entire approach. Are differences due to this fact?

Reply: We fully acknowledge that the four sites represent different processes. On the one hand we have two study sites, Gampenalm and Dreitorspitze, where the deposits of the talus slope are characterized by one event, while on the other sites, La Réunion and Zwieselbach valley, the slopes are formed by "continuous" rockfall activity. Rock avalanches have not taken place at our study sites. We find differences in deposition of boulders on the talus slopes regarding the block size and shape. For Gampenalm and Dreitorspitze, the size of the blocks does not play a major role. Here, the block shape acts as a moderating parameter. For the study sites of La Réunion and Zwieselbach valley, the block size influences the deposition on the talus slope while we could not find any influence of block shape on the influence of deposition. As we take the comment very seriously, we will also focus on the influence of the different process types and take it up in a more structured way in the manuscript.

**5. Conclusion**

Review Comment 32 – l357:

boulders with low axial ratios do also have a predominant rotation axis, as a perfect symmetric rock does not exist in nature. It rather has no large flat areas, hindering a re-acceleration after a landing on such a flat surface.

Reply: We agree with the reviewer that blocks with a lower axial ratio also have a rotation axis and that a symmetrical block does not exist in nature. However, we assume that flat and elongated blocks can also rotate. They can position themselves on their axis of rotation by moving downslope. In our analysis we come to the conclusion that for Gampenalm and Dreitorspitze blocks with a lower axial ratio achieve larger runout distances. In the case of Gampenalm, however, rounder blocks are deposited also in the upper part of the slope. In contrast, the parameter axial ratio does not play a role for the deposition of the study areas La Réunion and Zwieselbach valley.

---

## Author Comment (AC3) · 18 Dec 2020

**Response to reviewer comments on "Assessing the effect of lithological setting, block characteristic and slope topography on the runout length of rockfalls in the Alps and on the La Réunion island"**

**Kerstin Wegner, Florian Haas, Tobias Heckmann, Anne Mangeney, Virginie Durand, Nicolas Villeneuve, Philippe Kowalski, Aline Peltier, Michael Becht**

**nhess-2020-322**

**Response to Anonymous Referee #3 (RC3)**

We thank the anonymous referee #3 (RC3) for the constructive comments and detailed feedback. We greatly appreciate the suggested revisions which will improve the quality of our revised manuscript. In the following the original comments of the reviewer #3 are listed in blue and replied by us point-by-point in black.

The paper presents detailed analysis of rockfall deposits mapped using point cloud datasets captured at four sites with differing lithology. The analysis explores how the properties of rock shape and volume along with terrain roughness and morphology influence runout distance. The findings are discussed and contrasted with previous works on the topic. Differences are identified in the runout character according to lithological setting. A number of analysis methods fail to link the measured metrics with the potential rockfall runout process controlling them and perhaps could be enhanced by applying alternative metrics of rock shape and volume. Inconsistencies in terms applied to descriptions of rock shape lead to some confusions in understanding the findings regarding the influence of rock volume and shape on rockfall runout.

Reply: In the methods section (l.143) we have dealt with the description as well as the calculation of the block shape. We are aware that this can only be an approximation of both the block shape and the block size. When revising the manuscript, we will take care to use more consistent naming with regard to the block form. We decided to perform a more detailed analysis of a sample of selected blocks based on point-cloud data. This is expected to reveal the degree to which the metrics based on three axes represent the actual block shape and volume. Moreover, also in light of other reviewers' comments, we will conduct additional analyses on roughness (based on the original high-resolution point clouds) as a surface property (I) reflecting granulometry also smaller than the measured blocks and (II) influencing the runout length of rockfall particles along their calculated runout paths.

**The following observations and questions arise:**

Review Comment 1:

1) The differences in lithology between the study sites make this study of interest for rockfall. However, the fragmentation of rock during rockfall and the lithological control on preconditioning down slope fragmentation is poorly treated in this work. It would be interesting to consider the rock-mass properties of each of the rock walls. How do the site specific failure mechanisms, discontinuities and rock strength influence the distribution of rock fragments across the talus cone? Investigating this theme would be possible with the detail of data presented in this work, and have the potential to enhance the observations made in sections 4.1 and 4.2.

Reply: The aim of our analysis was not to analyse the fragmentation of the blocks during an event. We cannot do this with our data. Likewise, we cannot analyse the influence of lithology on fragmentation with our data because

structural data are not available at all sites. We can only measure the deposited blocks and put them into context with the respective lithology, as we have done. The intention of our analysis was not the analysis of different failure mechanisms, discontinuities and rock strength, but how the lithology influences the deposition on the talus cone. Even though we are of course aware that lithology also controls these factors. In the revision of the manuscript, we will make an effort to collect values from the literature for our study areas and different lithology (rock mass, strength) and add them to Table 1 (l102). The distribution of particles (of different sizes) on the talus cone will be analysed additionally by deriving the surface roughness from the high-resolution TLS point clouds. As roughness bears a relation to grain sizes (interesting specifically at sizes smaller than those we measured manually), we are confident to get a more complete idea of the depositional patterns.

Review Comment 2:

2) The use of rock axis ratios as a shape classification is an obvious choice given the volume of available data. Moreover, the geometric axis ratio has a bearing on the inertial axes of the rocks and its ability to maintain momentum during run out. The rock volume and shape are therefore coupled in their influence on runout behaviour. The analysis could benefit from coupling rock volume and shape.

Reply: The coupling of block size and block shape is mathematically difficult. While a multivariate regression of runout length on block shape AND volume failed, the identification of the most extreme block shapes (q10, q90, see Figure 7, l330) provided a clear indication that block shapes moderate the influence of volume (=>gravitational sorting). Regarding the mechanisms through which e.g. block shape may moderate runout length, we pledge to explain these more thoroughly in order to arrive at a more plausible interpretation of the findings.

**General comments to the text**
**1. Introduction**

Review Comment 3 – L34 to L38

The text introduces preconditioning and preparatory factors as well as triggering events. A clear distinction between preconditioning and triggering rockfalls for the examples of each that are given would be useful.

Reply: For La Réunion, it can be assumed that the rockfalls taking place there are mainly triggered by preconditioning factors such as the lithology, but also by the high seismic activity as trigger. For Gampenalm, Zwieselbach valley, and Dreitorspitze it cannot be determined exactly, but thawing of mountain permafrost can be excluded due to the altitude and exposure. Unfortunately, we do not have temperature and precipitation data.

Review Comment 4 – L39

"Due . . . importance", of what?

Reply: We wanted to express that due to past settlement development in mountain regions, the affected infrastructure can be partly affected by rockfall events that take place. We will rephrase this sentence in the revision of the manuscript.

Review Comment 5 – L61-L63

please clarify the use of proximal in this sentence, proximal to the source or deposits?

Reply: By "proximal" we mean the lower slope area where the blocks are deposited. This refers to the distance to the detachment area.

**2. Study sites**

Review Comment 6 – L97 & L 98

Please clarify the use of "untreated" with regards the rock mass description.

Reply: We will delete "untreated" in this context and refer only to "thick banked", as was done in the relevant lines (l97f).

Review Comment 7

Table 1 could benefit from details of the rock mass properties and strength.

Reply: We will search for the information on rock mass properties and strength for the study sites in the literature and add it to Table 1.

**3. Materials and Methods**

Review Comment 8 – L122

consider replacing "Both . . ." with "Each scanner works . . ..

Reply: We will follow the reviewer's comment.

Review Comment 9 – L124

clarity on the importance of coloured point clouds to the methodology would be useful.

Reply: We will deepen the topic of colouring point clouds and describe it in more detail.

Review Comment 10

Section 3.2 the method applied to obtain rock volume using the three principal geometric axes a b & c give the volume of a cuboid. To what extent does this method over estimate rock volume in this study?

Reply: We are aware that we probably overestimate the volumes of the blocks by our calculation. However, we note that we use the same approach for all areas, which in our view makes a comparison between the areas permissible. With our data, however, we are able to randomly quantify this overestimation. For each of the study sites, we will determine the volume in detail for a representative sample of ten blocks using the point cloud (with Riscan Pro) in order to be able to provide an estimate of the overestimation of the volumes.

Review Comment 11 – L152

The applied rock shape indicator bundles the platy and elongate rock shapes into the same indicator, please discuss this choice in more detail.

Reply: We will add more detailed information about the chosen method especially the block shape when revising the manuscript.

Review Comment 12 – L154

The axis ratio of 1 in which axes are the same length is referred to as a round shape. How does the applied shape classification account for the roundness of the rock? Could an axis ratio of 1 be described as "equant"? Please consider applying equant throughout the text. How do larger rocks with flat sides but axis ratio of 1 runout in comparison to a rounded rock?

Reply: We argue that the axial ratio does not precisely represent roundness. Where it approaches a value of 1, the block is thought to be a cuboid, whether for increasing axial ratio blocks become more irregular shaped (Here we refer to Glover et al., 2015.). Large blocks with flat sides and an axial ratio of 1: the corresponding blocks do not have flat sides, but their axes are almost the same length. In the revision of the manuscript, we will make sure to use a uniform naming of the block shape in the methods section as well as in the results and the conclusion in order to avoid misunderstandings.

Review Comment 13 – L168

"rock fall" please replace with rockfall.

Reply: We will follow the reviewer's comment.

**4. Results and Discussion**

Review Comment 14 – L190

please consider deleting "dispersion also".

Reply: We will follow the reviewer's comment.

Review Comment 15 – L193

please consider "clearly shown" in replacement of "well visible".

Reply: We will follow the reviewer's comment.

Review Comment 16 – L241

To what extent can this be attributed to fragmentation of rock during runout?

Reply: It is not impossible that fragmentation may occur during the fall. However, we cannot determine it with exact probability. For the corresponding study sites, we have to assume that the processes are continuous and not a single major event. In the revision, however, we will emphasise and explain that there is an uncertainty.

Review Comment 17 – L300

repetitive use of adverb "clearly" "obviously".

Reply: We will rephrase that in the revision.

**5. Conclusions**

Review Comment 18 – L345

the term sphericity is newly introduced in the conclusions, it is not clear how sphericity was measured in the methods. Is rounded or equant shape meant?

Reply: Sphericity or roundness are the wrong terms for the method used. This is our mistake. A measured block with an axial ratio of 1 is thought to be a cuboid. For this reason, it is possible to refer to equant. We will also take care to remain consistent in the manuscript with the naming of the terms while the revision. This also refers to your comments regarding line 152, line 154 and your introduction.

---

## Referee Report (RR1)

Comments to the authors of the revised manuscript under discussion entitled "*Assessing the effect of lithological setting, block characteristic and slope topography on the runout length of rockfalls in the Alps and on the La Réunion island*"

The presented manuscript deals with the four analysed talus slopes and a thorough mapping of lithological characteristics, being deposited rock shapes and sizes and its influence on runout length. The revised manuscript presents itself substantially improved and presents the complex findings in a more concise way.

I do thank the authors for their effort in revising the manuscript thoroughly, especially in the large effort in remaking the figures. IMHO it increases readability and significance and the revised manuscript is suitable for publication.

**Only technical remark**

Use of or ambiguity of rock shape and axial ratio: I am a bit confused by the statement " We used the block shape as a term to describe the axial ratio." (l159). Eq. 2 *is* the axial ratio and you term it also block shape (l 158).
Basically, the axis determine the axial ratio via Eq. 2 and thus describe the block shape (but are not able to catch the roundness of the block). Correct? Maybe rephrase this a little.

**Use of it in Table 4:**
How is the parameter block shape calculated for the numbers in the table? It is the "Axial ratio" according to Eq. (2) with b = 1? Maybe note that in the table title or change to Axial ratio for consistency reasons.